# Pesticide Residues in Fruits and Vegetables from Cape Verde: A Multi-Year Monitoring and Dietary Risk Assessment Study

**DOI:** 10.3390/foods14152639

**Published:** 2025-07-28

**Authors:** Andrea Acosta-Dacal, Ricardo Díaz-Díaz, Pablo Alonso-González, María del Mar Bernal-Suárez, Eva Parga-Dans, Lluis Serra-Majem, Adriana Ortiz-Andrellucchi, Manuel Zumbado, Edson Santos, Verena Furtado, Miriam Livramento, Dalila Silva, Octavio P. Luzardo

**Affiliations:** 1Toxicology Unit, Research Institute of Biomedical and Health Sciences (IUIBS), University of Las Palmas de Gran Canaria, Paseo Blas Cabrera s/n, 35016 Las Palmas de Gran Canaria, Spain; andrea.acosta@ulpgc.es (A.A.-D.); manuel.zumbado@ulpgc.es (M.Z.); 2Department of Environmental Analysis, Technological Institute of the Canary Islands, C/Los Cactus no 68, Polígono Industrial de Arinaga, 35118 Agüimes, Spain; rdiaz@itccanarias.org (R.D.-D.); mbernal@itccanarias.org (M.d.M.B.-S.); 3Social Sciences, Heritage and Food, Institute of Natural Products and Agrobiology (IPNA-CSIC), Av. Astrofisico Francisco Sánchez, 3, 38206 La Laguna, Spain; pablo.alonso.gonzalez@ipna.csic.es (P.A.-G.); eva.parga.dans@ipna.csic.es (E.P.-D.); 4Nutrition Research Group, Research Institute of Biomedical and Health Sciences (IUIBS), University of Las Palmas de Gran Canaria, Paseo Blas Cabrera s/n, 35016 Las Palmas de Gran Canaria, Spain; lluis.serra@ulpgc.es (L.S.-M.); adriana.ortiz@ulpgc.es (A.O.-A.); 5Spanish Biomedical Research Center in Physiopathology of Obesity and Nutrition (CIBERObn), 28029 Madrid, Spain; 6Entidade Reguladora Independente da Saúde (ERIS), Av. Cidade de Lisboa, 296-a, Praia 7600, Cape Verde; edson.santos@eris.cv (E.S.); verena.furtado@eris.cv (V.F.); miriam.livramento@eris.cv (M.L.); dalila.silva@eris.cv (D.S.)

**Keywords:** pesticide residues, fruits, vegetables, food safety, dietary exposure, risk assessment, hazard index, Cape Verde, monitoring program

## Abstract

Food safety concerns related to pesticide residues in fruits and vegetables have increased globally, particularly in regions where monitoring programs are scarce or inconsistent. This study provides the first multi-year evaluation of pesticide contamination and associated dietary risks in Cape Verde, an African island nation increasingly reliant on imported produce. A total of 570 samples of fruits and vegetables—both locally produced and imported—were collected from major markets across the country between 2017 and 2020 and analyzed using validated multiresidue methods based on gas chromatography coupled to Ion Trap mass spectrometry (GC-IT-MS/MS), and both gas and liquid chromatography coupled to triple quadrupole tandem mass spectrometry (GC-QqQ-MS/MS and LC-QqQ-MS/MS). Residues were detected in 63.9% of fruits and 13.2% of vegetables, with imported fruits showing the highest contamination levels and diversity of compounds. Although only one sample exceeded the maximum residue limits (MRLs) set by the European Union, 80 different active substances were quantified—many of them not authorized under the current EU pesticide residue legislation. Dietary exposure was estimated using median residue levels and real consumption data from the national nutrition survey (ENCAVE 2019), enabling a refined risk assessment based on actual consumption patterns. The cumulative hazard index for the adult population was 0.416, below the toxicological threshold of concern. However, when adjusted for children aged 6–11 years—taking into account body weight and relative consumption—the cumulative index approached 1.0, suggesting a potential health risk for this vulnerable group. A limited number of compounds, including omethoate, oxamyl, imazalil, and dithiocarbamates, accounted for most of the risk. Many are banned or heavily restricted in the EU, highlighting regulatory asymmetries in global food trade. These findings underscore the urgent need for strengthened residue monitoring in Cape Verde, particularly for imported products, and support the adoption of risk-based food safety policies that consider population-specific vulnerabilities and mixture effects. The methodological framework used here can serve as a model for other low-resource countries seeking to integrate analytical data with dietary exposure in a One Health context.

## 1. Introduction

The growing use of pesticides worldwide in the search for increasing agricultural yields and avoiding losses due to plant pests has led to increasing consumer awareness and concern about the presence of pesticides in foods [1]. Developed regions such as the European Union, the United States and Japan rank among the top pesticide consumers globally [2]. However, in these regions, the risks associated with pesticide exposure are mitigated through the implementation of comprehensive residue monitoring programs, which provide essential data for regulatory risk assessment and enforcement [3,4]. In developing countries, however, both the inappropriate use of pesticides and the lack of monitoring strategies are important food safety concerns [5]. In Africa in particular, the pesticide management capacities vary significantly between countries [6,7,8,9].

Developing countries often test fresh produce intended for the export market to guarantee compliance with standards set by the destination country, but no monitoring strategies target both local and imported produce at sale in the market [7]. This is significant given that food intake is generally the dominant route of pesticide exposure in the general population [10,11]. In this context, Cape Verde’s inclusion in the PERVEMAC I and II projects provided a unique opportunity to generate representative data on the contamination of plant-based foods available to local consumers [12,13]. PERVEMAC has implemented an unprecedented monitoring program for pesticide and mycotoxin residues in fruits, vegetables, and cereals consumed in the Macaronesian archipelagos, which share Atlantic climates, volcanic soils, and significant agri-livestock sectors oriented toward local and export markets [14,15]. The climate in these regions promotes intensive pesticide use. For instance, the Canary Islands present the highest consumption of pesticides per hectare of all Spanish regions [16].

Cape Verde is an archipelago comprising nine inhabited islands located on the West African coast, 500 km away from Senegal. Its 561,901 inhabitants are concentrated in the islands of Santiago (260,000 inhabitants) and São Vicente (76,000 inhabitants). Despite the difficult arid conditions, worsened by drought in recent years, Cape Verde still preserves 11.2% of the territory devoted to agriculture [17]. The main crops are sugar cane, banana and mango, with extensive self-consumption gardening in rural areas [18]. Aside from drought, pest attacks on fruit and vegetables are the main drivers of food insecurity in the archipelago [12]. This is significant, given that almost 32% of the population are under-nourished and 20% of rural families are in conditions of food insecurity according to FAO [19,20]. The population of Cape Verde is in a process of transition towards a Western food paradigm dominated by refined carbohydrates, high fats and sugar, with a decreasing consumption of fruits and vegetables and a growing prevalence of obesity [18].

This study focuses on fruits and vegetables, which remain dietary staples in Cape Verde, providing key nutrients and vitamins. Most vegetables consumed are locally grown, while imported fruits—mainly from Europe—include varieties not produced domestically, such as apples, pears, oranges, lemons, kiwis, and watermelons. Despite their health benefits, concerns are rising about pesticide residues, which can originate from direct application as well as from contaminated soil and irrigation water [6]. In Cape Verde, self-sufficient farmers subsist in rural areas, but most urban dwellers depend on commercial markets for their fresh produce. Most of these fruits and vegetables may contain pesticide residues causing negative effects for the human health [21,22].

Pesticides are xenobiotic and therefore, from a public health and environmental perspective, there is a need to monitor their use and presence in the final produce [23]. Such monitoring strategies have been implemented for decades in most developed countries [15]. They generally include the study of the residue levels, potential risk for consumers, and compliance with maximum residue limits (MRLs). Reference MRLs are set internationally by the Codex Alimentarius [24] and regionally by authorities such as the European Commission [25]. However, developing countries such as Cape Verde lack monitoring strategies and therefore very limited data are available on pesticide residues in fruits and vegetables, and consequently on the dietary intake of pesticides by the population. Previous studies focusing on Cape Verde have analyzed cereals, the sustainable use of pesticides and explored the presence of pesticide residues in humans through biomonitoring strategies [12,13,18,26,27]. However, no data had been reported previously on pesticide residues in fruits and vegetables in Cape Verde from a multi-year perspective. This study aimed to monitor pesticide residues in fruits and vegetables marketed in Cape Verde, with the dual objective of evaluating compliance with international safety standards and conducting a dietary risk assessment.

## 2. Materials and Methods

### 2.1. Study Area and Sample Selection

Cape Verde is a volcanic archipelago located off the coast of West Africa, composed of ten islands, nine of which are inhabited. With a population of approximately 500,000 inhabitants, the country faces structural limitations in agricultural production due to arid climatic conditions, scarce freshwater resources, and limited arable land. As a result, Cape Verde relies heavily on food imports to meet the nutritional needs of its population, particularly for fruits and vegetables, which are only partially supplied through domestic production.

The present study was conducted within the framework of the European cooperation projects PERVEMAC I (2017–2018) and PERVEMAC II (2019–2020), aimed at assessing chemical contaminants in food across Macaronesian territories [12,13,28,29]. In Cape Verde, the pesticide residue monitoring component was implemented in collaboration with national regulatory bodies: the Agência de Regulação e Supervisão dos Produtos Farmacêuticos e Alimentares (ARFA) during the first edition of the project, and the Entidade Reguladora Independente da Saúde (ERIS) during the second.

Sampling was carried out in five of the country’s most populated islands—Santo Antão, São Vicente, Santiago, Boa Vista, and Fogo—which together represent a broad cross-section of Cape Verde’s food retail and consumption dynamics. The selection of food commodities and the number of samples per item were determined by ARFA and ERIS based on their availability and relative consumption levels in the local population, and taking into account the framework of the multi-annual coordinated European Union monitoring programme for pesticide residues in food of plant origin. A total of 570 samples were collected from open markets, municipal stalls, and supermarkets, comprising both locally produced and imported fruits and vegetables.

Sampling took place approximately every four months between 2017 and 2020, ensuring coverage of all seasons across the multi-year period. However, seasonal stratification was not applied in the analysis due to the limited number of samples per item, in order to preserve statistical robustness in exposure estimates. In the first sampling period (2017–2018), 383 samples were collected, distributed across 27 types of fresh produce. The most frequently sampled items included tomato (n = 74), cabbage (n = 17), orange (n = 14), papaya (n = 14), apple (n = 16), and carrot (n = 25). During the second period (2019–2020), 187 additional samples were collected, again focusing on the most commonly consumed fruits and vegetables in the local diet. The most sampled items during this phase were again tomato (n = 63), cabbage (n = 17), pear (n = 11), and carrot (n = 14). Table 1 summarizes the sample counts per food item in each sampling phase (Table 2).

### 2.2. Sample Preparation and Residue Extraction

Pesticide residue analysis was carried out at the Residues Laboratory of the Environmental Analysis Department of the Instituto Tecnológico de Canarias, S. A. (ITC), designated by the Canarian Government within the framework of official controls on pesticide residues, and accredited by the Spanish National Accreditation Body (ENAC), according to the requirements in UNE-EN ISO/IEC 17025 [30], for most of used pesticides multiresidues methods. As accepted by all Member States of European Union, Document SANTE [31,32], Analytical Quality Control and Method Validation Procedures for Pesticide Residue Analysis in Food and Feed, was adopted as technical guideline for the in-house methods validated. The laboratory employs an internationally validated multiresidue extraction method based on the QuEChERS protocol, in accordance with the Association of Official Analytical Chemists (AOAC) guidelines.

For each sample, 15 g of homogenized plant material were weighed into a 50 mL centrifuge tube. In the case of aromatic herbs, the sample was prepared by mixing 7.5 g of plant material with 7.5 g of ultrapure water to account for matrix-specific extraction differences. A mixture of internal standards (triphenyl phosphate at 10 µg/mL and chlorpyrifos-D10 at 5 µg/mL) was added and allowed to equilibrate for 15 min. Then, 15 mL of acetonitrile containing 1% glacial acetic acid were added, and the sample was vigorously shaken, followed by mechanical agitation using a Agitax™ (AGYTAX LAB, Madrid, Spain) programmable multi-tube vortex system.

Subsequently, a commercially prepared extraction salt packet (containing 6 g of MgSO_4_ and 1.5 g of sodium acetate) was added to induce phase separation. After agitation and centrifugation (3900 rpm for 10 min), a clean-up step was performed using 8 mL of extract and a dispersive solid-phase extraction (d-SPE) cartridge filled with 1.2 g MgSO_4_ and 0.4 g PSA (primary secondary amine). The resulting supernatant was centrifuged again and evaporated under reduced pressure using a centrifugal evaporator or rotavapor at 35–40 °C.

The dry residue was reconstituted in 200 µL of acetone and diluted with 1.8 mL of cyclohexane, regardless of whether the extract was intended for GC-MS or LC-MS/MS analysis. In all cases, the final solution was filtered through a 0.2 µm PTFE syringe filter into a 2 mL chromatographic vial. If the extract appeared turbid, approximately 150 mg of anhydrous MgSO_4_ were added directly into the syringe prior to filtration to ensure clarity.

This extraction method was validated for a broad spectrum of pesticide classes and is routinely used in official monitoring programs for compliance with Codex Alimentarius and European Union residue standards. Recovery, sensitivity, and repeatability parameters met the required criteria for reliable multiresidue detection in fruit and vegetable matrices.

For the analysis of dithiocarbamates, a specific extraction method was used based on acid-reduction digestion adapted to their chemical characteristics. Four grams of homogenized sample were reacted with 10 mL of a solution containing 3 g of tin(II) chloride in 37% HCl and 4 mL of isooctane to trap the released carbon disulfide (CS_2_). The mixture was sonicated for 10 min, heated at 90 °C for 90 min, cooled, sonicated again briefly, and stored at 4 °C before analysis. An aliquot of the isooctane phase was analyzed by gas chromatography with pulsed flame photometric detection. Results were expressed as total dithiocarbamates (CS_2_ equivalents).

### 2.3. Multi-Residue Analysis by Chromatography

All pesticide residue analyses were conducted at the Pesticide Residue Laboratory of the Instituto Tecnológico de Canarias (ITC), which is accredited by the Spanish National Accreditation Body (ENAC, Entidad Nacional de Acreditación) under the UNE-EN ISO/IEC 17025 standard. The laboratory holds an open scope accreditation for the determination of pesticide residues in fruits and vegetables using GC-MS/MS and LC-MS/MS techniques. This accreditation ensures compliance with internationally recognized quality standards, including regular audits, method validation, and performance criteria established in the SANTE/11312/2021 guidance document. The chromatographic technique applied depended on the period of analysis, due to the progressive upgrade of analytical instrumentation during the study timeline.

Between 2017 and 2018, samples were analyzed using two complementary techniques: gas chromatography–mass spectrometry with ion trap (GC-IT-MS/MS) and liquid chromatography–tandem mass spectrometry (LC-MS/MS). The LC-MS/MS analysis was conducted on a Varian 320 MS triple quadrupole system equipped with two Varian 212-LC pumps and a Varian 410 autosampler. Chromatographic separation was achieved using an ACE 3 C18-AR column (100 × 2.1 mm, 3 µm particle size) maintained at 40 °C. The mobile phases consisted of 5 mM ammonium formate with 0.2% formic acid (solvent A) and methanol:solvent A (90:10, *v*/*v*) as solvent B. The injection volume was 10 µL, with electrospray ionization operated in positive mode (ESI+), using a spray voltage of 5000 V and desolvation temperature of 200 °C.

In parallel, GC-MS/MS analyses were carried out using a Varian 3800 gas chromatograph coupled to a Varian 4000 ion trap mass spectrometer and a Varian 8400 autosampler. The separation was performed on an SGE BPX-5 column (30 m × 0.25 mm, 0.25 µm). The oven temperature program consisted of an initial hold at 70 °C for 3.5 min, followed by a ramp of 25 °C/min to 180 °C (held for 10 min), and a final ramp of 4 °C/min to 300 °C (held for 10 min). Injections were made using large volume injection (10 µL) in a Varian 1079 programmable temperature vaporizer (PTV)(Varian Inc., Palo Alto, USA) injector. Ion trap MS parameters included transfer line, manifold, and trap temperatures of 280 °C, 50 °C, and 220 °C, respectively.

Starting in 2019, all multiresidue analyses were performed exclusively by GC-MS/MS using a more advanced system: an Agilent 7890B gas chromatograph coupled to an Agilent 7000D triple quadrupole mass spectrometer, with an Agilent 7693A autosampler. Chromatographic separation was achieved using two HP-5ms columns (15 m × 0.25 mm ID, 0.25 µm) arranged in backflush configuration. The oven program began at 60 °C (1 min hold), followed by a ramp of 40 °C/min to 170 °C and then 10 °C/min to 310 °C (3 min hold). Injections (2 µL) were performed in split/splitless mode using an Agilent Ulti Inert inlet held at 280 °C. The ion source, transfer line, and quadrupole temperatures were 280 °C, 280 °C, and 150 °C, respectively.

This equipment renewal enhanced detection sensitivity but slightly reduced the overall analytical scope. While up to 180 active substances could be analyzed using the initial dual-method platform (GC-IT-MS/MS and LC-MS/MS), the upgraded GC-MS/MS method applied from 2019 onwards allowed for the detection of 142 compounds. All methods operated in multiple reaction monitoring (MRM) mode and were validated for use in official control programs in compliance with European regulatory standards.

Dithiocarbamate analysis was performed separately, based on the generation and quantification of carbon disulfide (CS_2_) through a gas chromatography quantitative method. Following sample extraction and acid digestion with tin(II) chloride and hydrochloric acid, the liberated CS_2_ was quantified by gas chromatography with flame photometric detection (GC-FPD), using a Varian 3800 system equipped with an CP-Sil5 for sulphur capillary column (30 m × 0.32 mm, 4.0 µm). The injector and detector were maintained at 200 °C and 300 °C, respectively. Helium was used as the carrier gas at a flow rate of 2 mL/min. Quantification was based on external calibration using certified CS_2_ standard as the reference compound, and results were expressed as total dithiocarbamates in µg/kg.

Quality control (QC) procedures were applied systematically during the analysis. Each analytical batch included blank samples, spiked samples at relevant concentration levels, and quality control standards. These QC measures followed the recommendations of the SANTE/12682/2019 guideline, ensuring the reliability, precision, and accuracy of the analytical results.

The full list of compound-specific MRM transitions, retention times, and corresponding validation parameters is provided in Appendix A, and in Appendix A.

### 2.4. Estimation of Dietary Exposure and Risk Assessment

The dietary exposure assessment was conducted using a deterministic approach based on the combination of median residue concentrations measured in this study and average daily food consumption figures obtained from the national nutrition survey ENCAVE 2019 (www.eris.cv). This survey collects data at the household level, with a single respondent (typically the female head of household) reporting on behalf of all members. Given the relatively homogeneous dietary patterns observed in Cape Verdean households—particularly in rural areas—and the absence of individual-level variability data, probabilistic modeling methods such as Monte Carlo simulation could not be applied. To account for vulnerable subgroups, a complementary risk assessment was conducted for children by adjusting consumption and body weight to estimate intake per kilogram.

For each active substance and food item, the estimated daily intake (EDI) was calculated by multiplying the median residue level (µg/g) by the average daily consumption of that item (g/person/day). The total exposure per compound was then obtained by summing contributions from all relevant food sources. Calculations were performed separately for fruits and vegetables and expressed in micrograms per person per day (µg/person/day).

In samples with pesticide concentrations below the LOQ, a single value randomly selected between zero and the LOQ was imputed at the outset of the analysis and kept fixed throughout all calculations. This procedure, while involving an element of random selection, was applied deterministically and consistently across the dataset. The goal was to avoid artificial clustering of data at fixed substitution points (e.g., LOQ/2), thereby preserving data variability without relying on probabilistic simulation. This approach reduces bias and preserves the distributional characteristics of the dataset, particularly in scenarios with high censoring and non-normal distributions, and is supported by previous statistical evaluations of left-censored environmental data [33,34].

To assess toxicological risk, the EDI of each active substance was divided by its Acceptable Daily Intake (ADI) as established in the European Union Pesticide Database [35]. These values, expressed in milligrams per kilogram of body weight per day (mg/kg bw/day), were adjusted assuming a default adult body weight of 70 kg. The resulting values were expressed as hazard indices (HIs), allowing direct comparison with the risk threshold (HI = 1). A HI value below 1 indicates that chronic exposure remains within the safety margin.

A cumulative risk assessment was also conducted by summing the individual hazard indices across pesticide classes—namely insecticides, fungicides, herbicides, and acaricides—as well as across all compounds. This approach, commonly used in regulatory practice, assumes additive effects within and across chemical groups [1]. Although this assumption simplifies the toxicological interpretation, it may underestimate or overestimate actual risk due to potential synergistic, antagonistic, or overlapping modes of action among substances. The cumulative hazard index (HI) was calculated as the sum of the ratios between the EDI and the ADI for all detected substances, according to the following equation:HI = Σ (EDIᵢ/ADIᵢ) where i denotes each pesticide detected in the sample group. This method follows international guidelines for cumulative dietary risk assessment.

Additionally, a child-specific assessment was performed for the most vulnerable age group (6–11 years), based on adjusted consumption and body weight parameters. Estimated consumption ratios relative to adults were derived from ENCAVE 2019 data, and an average body weight of 25 kg was assumed for children. This allowed the calculation of age-specific hazard indices and the identification of subpopulation risks not captured in the adult-centered model.

### 2.5. Statistical Analysis

All statistical analyses were performed using GraphPad Prism v10.0 (GraphPad Software, Boston, USA). The distribution of pesticide residue concentrations was assessed using the Kolmogorov–Smirnov test. As most variables did not follow a normal distribution, results are presented as mean ± standard deviation (SD), median, and range to adequately describe central tendency and dispersion.

Comparisons between categorical groups—such as local vs. imported produce or fruits vs. vegetables—were evaluated using the non-parametric Mann–Whitney U test. A two-tailed *p*-value below 0.05 was considered indicative of statistical significance.

## 3. Results

### 3.1. Occurrence of Pesticide Residues in Fruits and Vegetables in Cape Verdean Markets

A total of 570 samples of fruits and vegetables were analyzed for pesticide residues in retail markets across Cape Verde. Among these, 191 were fruit samples and 379 were vegetables. Residues of pesticides were quantified in 122 fruit samples (63.87%) and 50 vegetable samples (13.2%), revealing a markedly higher contamination frequency in fruits than in vegetables.

In total, 80 different active substances were quantified, including insecticides, fungicides, acaricides, and herbicides. Altogether, 636 pesticide residues were detected in fruits (3.33 residues/fruit sample), corresponding to 75 different active substances, and 337 residues were found in vegetables (0.89 residues/vegetable sample), involving 38 different substances. The average number of residues per sample was therefore 3.7 times higher in fruits than in vegetables.

Despite the high detection rate and diversity of residues, only one sample raised potential regulatory concern: a clementine with over 3000 μg/kg of propiconazole—a fungicide that was still authorized in the European Union at the time of sampling, with an MRL set at 5 mg/kg. Although this MRL was later reduced to 0.01 mg/kg following the ban of propiconazole in December 2018, the observed level did not constitute a legal violation when the sample was collected.

Figure 1 presents the distribution of pesticide classes across both groups. In both fruits and vegetables, insecticides were the most frequently detected class. In the case of vegetables, the number of different insecticides and fungicides was similar. Notably, no acaricides were detected in vegetables.

Among fruits, the highest number of residues was detected in grape (n = 27), pear (n = 21), papaya (n = 20), and orange (n = 21). Up to 18 different pesticide residues were found in individual fruit types. Clementines and lemons exhibited the highest total median concentrations, with values reaching 2537 µg/kg and 1767 µg/kg, respectively. However, individual measurements revealed substantial variability. For instance, fludioxonil in lemon reached the value of 2710 µg/kg, while pyrimethanil in clementine reached 660 µg/kg. These pesticides are used post-harvest on citrus fruits, which is the reason for the high concentrations found and the MRLs established for these fungicides.

These values are presented in Table 1, which reports median concentrations, and further details including means, standard deviations, and concentration ranges (min–max) are provided in Appendix A.

Conversely, fruits with the lowest overall pesticide concentrations included watermelon, prune, and kiwi, all of which showed low total medians and a limited number of detected substances. Notably, no residues were detected in any of the mango (n = 11) or strawberry (n = 6) samples. These two products are either locally produced (mango) or predominantly sourced from local growers (strawberry). No post-harvest fungicides are used on these fruits.

Regarding vegetables, pesticide residues were detected in 11 out of the 12 types analyzed. The high number of tomato samples primarily reflects the strong domestic demand for this crop, which is widely cultivated and consumed throughout Cape Verde. Tomato (n = 134) was by far the most frequently sampled crop, followed by red cabbage (n = 41) and cabbage (n = 34). These three also ranked among the most contaminated vegetables. Red cabbage showed the highest total median concentration (202.2 µg/kg), followed by cabbage (124.7 µg/kg) and pepper (130.0 µg/kg). In contrast, the lowest contamination levels were observed in lettuce (4.8 µg/kg), parsley (8.2 µg/kg), and zucchini (11.7 µg/kg). A full summary of median concentrations is presented in Table 2, while additional statistics including mean, standard deviation, and concentration ranges are detailed in Appendix A.

Among the most frequent residues in vegetables were dithiocarbamates, with particularly high median values (up to 200 µg/kg in tomato). Other commonly detected compounds included fungicides like fluopyram, boscalid, and various triazoles, as well as insecticides such as deltamethrin, dimethoate, cypermethrin, and imidacloprid. It is important to note that the analytical determination of dithiocarbamates refers to a broad chemical group that includes various active substances (e.g., mancozeb, metiram, propineb, thiram, ziram). Some of these were currently authorized under European Union standards, while others were already banned during the sampling period of the present effort. Since the analytical method used does not distinguish between these compounds, it is not possible to assess legal compliance based solely on the measured concentrations.

The high number of tomato samples reflects the strategic importance of this crop in Cape Verdean agriculture. Tomato is one of the few irrigated crops and is cultivated both for domestic consumption and export, particularly to the European Union (Portugal). Consequently, monitoring pesticide residues in tomato is especially relevant for ensuring regulatory compliance. Figure 2 summarizes the number of samples that exceeded the maximum residue limits (MRLs) established either by the Codex Alimentarius—an international reference adopted by countries without national standards—or European legislation, for both fruits and vegetables. A total of 22 non-compliant samples were identified under Codex standards, and 48 under EU legislation, highlighting the need for harmonization and stricter control for products with export potential. Notably, of the 108 pesticide residues detected in tomato samples, 56 corresponded to active substances not currently authorized to be used in the European Union. However, EU legislation may tolerate certain residues in imports from countries regulated under Codex standards.

The international reference legislation for establishing MRLs for pesticide residues in fruits and vegetables is the Codex Alimentarius. However, the Codex does not cover all possible pesticide/plant commodity combinations. In these cases, countries have two options: develop their own legislation or use existing legislation as a reference. The two most relevant ones are those established by the Environmental Protection Agency in the United States and European legislation. Cape Verde uses European legislation as a reference when it is not possible to apply the Codex.

Figure 3 shows the distribution of the number of pesticide residues detected per sample in fruits and vegetables, using boxplots. For fruits, samples are stratified by origin (imported vs. locally produced), while for vegetables only local production is represented, as only six imported samples were available, reflecting the structure of the Cape Verdean horticultural market.

The data reveal a clear pattern: fruits tend to contain more residues than vegetables, and imported fruits contain significantly more residues per sample than their locally produced counterparts. The median number of residues was 8 for imported fruits, 0 for locally produced fruits, and 1 for vegetables. The distribution in imported fruits also exhibited a broader range, with some samples containing up to 22 different residues. In contrast, locally produced fruits and vegetables showed narrower distributions and lower counts, suggesting fewer contaminants overall.

Figure 4 further supports these findings by comparing the total concentrations of pesticide residues (ng/g) in fruits and vegetables across pesticide classes. Fruits showed significantly higher total concentrations of acaricides, fungicides, and insecticides than vegetables (*p* < 0.0001), as well as a higher overall pesticide burden (*p* = 0.0092). No significant difference was observed for herbicides. These results confirm that, beyond containing more individual residues, fruits also exhibit higher total contamination loads across most pesticide categories.

### 3.2. Dietary Exposure to Pesticide Residues in the Cape Verdean Population

To assess the dietary exposure of the adult population in Cape Verde to pesticide residues, we combined the median concentration values for each active substance in each food item (as determined in this study) with the average daily consumption of fruits or vegetables reported in the ENCAVE 2019 nutritional survey. This approach provides an estimate of chronic exposure in micrograms per person per day (µg/person/day), specific to each compound and food source. The food consumption data, obtained from a representative sample of the population of Santiago Island, revealed that the average daily consumption of fruits was 165.8 g/person and that of vegetables was 186.5 g/person.

Among the 80 active substances detected, 54 were associated with measurable dietary exposure from fruits and/or vegetables. The estimated total daily exposure to all pesticide residues combined was 22,154 µg/person/day, with fruits accounting for the majority of the intake (20,449 µg/person/day) and vegetables contributing a smaller portion (1705 µg/person/day).

A small number of active substances accounted for most of the exposure burden. Specifically, propiconazole, a fungicide detected primarily in citrus fruits, represented the largest contributor to dietary exposure, with an estimated intake of 15,076 µg/person/day. This value was mainly driven by a single sample of clementine containing more than 3000 µg/kg of this compound. Other significant contributors included fludioxonil (1346 µg/person/day), chlorpyrifos (642 µg/person/day), and spirodiclofen (447 µg/person/day).

Overall, exposure through vegetables was markedly lower. The highest contributions from vegetables were linked to dithiocarbamates, fluopyram, boscalid, and triazole fungicides, but in all cases the individual estimated intakes were well below 300 µg/person/day.

These estimates provide a detailed overview of dietary exposure to pesticide residues based on real consumption patterns and contamination levels in local markets. A complete list of estimated daily intakes for all active substances and food groups is presented in Table 3, with expanded calculations available in Appendix A.

### 3.3. Risk Characterization of Dietary Exposure to Pesticide Residues

To evaluate the potential health risks associated with chronic dietary exposure to pesticide residues in Cape Verde, hazard indices (HIs) were calculated for each active substance by dividing the estimated daily intake (EDI, expressed in µg/person/day) by the corresponding Acceptable Daily Intake (ADI) values published by the European Union. These values were then normalized to body weight (mg/kg bw/day) and aggregated to generate class-specific and total hazard indices. A hazard index ≥ 1.0 was used as the reference threshold for potential toxicological concern.

Figure 5 presents the cumulative hazard indices, disaggregated by pesticide class (acaricides, fungicides, herbicides, and insecticides) and by food group (fruits vs. vegetables). For the adult population, the total cumulative hazard index reached 0.416. Fruits contributed the majority of the toxicological burden (0.270), nearly double that of vegetables (0.146). Among pesticide classes, insecticides posed the greatest risk (0.283), followed by fungicides (0.124), while acaricides and herbicides contributed minimally (0.0088 and 0.0004, respectively). These results align with the contamination profiles described earlier, in which fruits—particularly imported ones—were associated with a greater number and diversity of pesticide residues.

Importantly, the same assessment was extended to children aged 6–11 years, a population group known for its increased vulnerability to chemical exposures. Since the ENCAVE 2019 nutritional survey does not provide disaggregated consumption data by individual food items for this age group, the exposure estimation in children was approached indirectly. Specifically, total consumption of fruits and vegetables for children was obtained from ENCAVE, and proportional consumption relative to adults was calculated: 61% for vegetables and 91% for fruits. These proportions were then applied to the adult food item–specific data, and the results were adjusted to reflect a lower average body weight in children (25 kg).

As shown in the lower panel of Figure 5 the cumulative hazard index for children approached the threshold of 1.0, indicating a level of concern under current dietary and contamination conditions. While adult exposure levels remained within regulatory safety margins, the much narrower safety buffer observed in children raises concern, especially considering their ongoing development, higher intake relative to body weight, and potential for long-term cumulative effects.

This child-specific estimate underscores the limitations of adult-centric models in food safety evaluation and highlights the importance of incorporating population-specific dietary and physiological parameters when assessing chronic exposure to pesticide residues.

## 4. Discussion

### 4.1. Prevalence and Patterns of Pesticide Contamination in Fruits and Vegetables of the Cape Verdean Market

The detection of pesticide residues in nearly two-thirds of fruit samples (63.9%) and in a significantly smaller proportion of vegetables (13.2%) reveals a marked asymmetry in contamination levels across food groups. Similar trends have been reported in other West African countries, such as Ghana and Cameroon, where fruits are frequently identified as high-risk commodities due to the prevalence of residues above legal limits or the use of unauthorized compounds [6,24,36]. These patterns are often linked to structural shortcomings in regulation, training, and the circulation of counterfeit or obsolete pesticides [25].

In Cape Verde, however, this disparity is compounded by the fact that the most heavily contaminated products were predominantly imported fruits, mostly from European markets or third countries with preferential trade agreements. Although these imports are often compliant with Codex Alimentarius standards, many of the detected substances would not be permitted under current European Union regulations, particularly for goods intended for intra-EU trade [3,37]. This paradox—where food not eligible for EU markets can still be exported under more permissive international rules—highlights the regulatory asymmetries embedded in global food trade and the vulnerability of small, import-dependent nations like Cape Verde, which rely on less protective international benchmarks.

The wide range of pesticide residues observed—80 different active substances in total—surpasses the diversity typically reported in European monitoring programs [1,4,21,38]. This heterogeneity reflects not only a broad spectrum of crop protection practices but also inconsistencies in the harmonization of international residue legislation. The detected compounds included all major pesticide classes, with insecticides being the most frequent, in line with usage patterns documented in tropical agriculture. However, the notable presence of fungicides and herbicides—especially in fruits—suggests additional sources of contamination related to post-harvest storage and preservation [6,9].

An analysis of the distribution of residues per sample further illustrates these patterns. Imported fruits showed the highest contamination profiles, with a median of 8 residues per sample, compared to 0 in locally produced fruits and 1 in vegetables. This is consistent with reports from other African markets, where imported produce—despite being formally regulated—tends to exhibit more complex residue profiles due to the intensity of pesticide applications throughout the production and logistics chain [5,10]. In this study, up to 22 different residues were found in a single fruit sample, suggesting overlapping treatments at multiple stages.

The frequent detection of compounds such as chlorpyrifos, profenofos, and dithiocarbamates—some of which are banned in the EU or lack Codex MRLs for specific crops—raises important concerns regarding international compliance and consumer safety [8]. Moreover, the analytical method used for dithiocarbamates (based on CS_2_ release) cannot distinguish between individual compounds, which means that both authorized substances (e.g., mancozeb) and banned ones (e.g., ziram, propineb) may be indistinctly reported. This analytical limitation underscores the need to integrate legal frameworks with more compound-specific detection techniques, especially when assessing compliance in international trade.

Equally important is the contrast between the relatively clean residue profiles of locally produced food and the elevated contamination levels in imports. While local agriculture currently supplies only a modest share of the national food basket, it shows a consistently lower pesticide burden. This represents a potential leverage point for national food policy: the promotion of sustainable local production, supported by training, access to low-risk inputs, and market incentives, could enhance food safety and reduce dependence on high-risk imports [17]. At the same time, the data highlight the urgent need to reinforce import controls and progressively align national residue standards with more protective international benchmarks, both to safeguard consumer health and to maintain Cape Verde’s credibility as an emerging agricultural exporter.

Taken together, these findings provide a comprehensive overview of the occurrence and origin of pesticide residues in food sold in Cape Verde. Beyond serving as a baseline for regulatory and policy efforts, they raise important questions about the potential health implications of chronic dietary exposure, particularly through imported fruits. The following section will therefore assess the extent to which the detected residues may contribute to population-level exposure and explore whether this poses a health concern under current toxicological and regulatory thresholds.

### 4.2. Dietary Exposure and Risk Characterization

The estimation of chronic dietary exposure to pesticide residues in Cape Verde reveals a relatively modest cumulative risk under current consumption patterns. When exposure levels were compared to the Acceptable Daily Intake (ADI) established by the European Union for each active substance, the resulting hazard index for the adult population remained well below the toxicological threshold of concern (HI = 1). The overall cumulative index was 0.416, with fruits contributing disproportionately (0.270) compared to vegetables (0.146), in line with their higher contamination profiles. It is important to note that cumulative hazard indices, while widely used and accepted in regulatory frameworks, do not account for potential synergistic or antagonistic interactions among compounds. More refined approaches based on toxicokinetic and toxicodynamic modeling require detailed experimental data not currently available for most of the substances included in this study. Our results suggest a moderate toxicological burden associated with routine dietary intake—well below acute risk levels, but not negligible in a One Health framework.

Of particular concern is the extrapolation of these data to the most vulnerable segment of the population: children. By adjusting consumption data and body weight to reflect the intake patterns of children aged 6–11 years (average body weight of 25 kg), and accounting for their comparatively higher consumption of fruits (91%) and vegetables (61%) relative to adults, we estimated that the cumulative hazard index in this group nearly reaches the safety threshold for toxicological concern. These results suggest that children may be exposed to chronic pesticide levels that pose a potential health risk under current dietary and contamination scenarios. While the adult index remains within regulatory margins, the child-specific estimate illustrates the limitations of using adult-centric models in food safety evaluation and underscores the need for population-specific assessments. The heightened vulnerability of children to chemical exposures—due to their physiological immaturity, developing organ systems, and higher intake per kilogram of body weight—is well established in the toxicological literature and necessitates stricter protective measures [4,39,40].

It is important to note that all hazard indices discussed below refer to adult exposure. For children, these values could be more than twice as high due to their greater relative intake and lower body weight. Therefore, the substances identified as major contributors to risk in adults are likely to exert an even more significant impact in children.

The total risk, as calculated, assumes purely additive effects between substances. While this is a standard approach in dietary risk assessment, it does not account for potential interactions among compounds. In reality, certain pesticide mixtures may act synergistically, potentiating each other’s toxic effects even at doses individually below concern thresholds. Conversely, some may interact antagonistically or compete for metabolic pathways, reducing the overall toxic potential. The absence of mixture-specific toxicological data introduces uncertainty in the cumulative risk assessment and underscores the need for future studies that incorporate mechanistic or experimental evidence on mixture effects [3,37,41,42].

Among the active substances with measurable dietary exposure, a limited number of compounds accounted for the majority of the risk burden. Omethoate, an organophosphate insecticide and known cholinesterase inhibitor, emerged as the leading contributor to total risk, with a hazard index of 0.071. Although banned in the EU, its presence in imported products illustrates gaps in harmonization between international and European standards. Oxamyl, another highly toxic carbamate, contributed 0.050 to the total index despite its relatively low concentration, reflecting its extremely low ADI. Similarly, imazalil, a fungicide commonly used in post-harvest treatments of citrus fruits, accounted for 0.041, highlighting the relevance of storage and transport processes as exposure sources.

Notably, dithiocarbamates, detected as a group and widely used in tropical agriculture, added 0.022 to the cumulative index. The inability to resolve individual compounds within this group—some of which are banned—adds a layer of uncertainty to the compliance and risk narrative. Iprodione, another fungicide primarily associated with imported fruits, also surpassed the 0.02 threshold, with an index of 0.021.

Beyond these top contributors, several other compounds approached or exceeded a hazard index of 0.01, collectively accounting for a substantial fraction of the total burden. Among them, cypermethrin and lambda-cyhalothrin, two pyrethroid insecticides, contributed nearly 0.019 each. Both are neurotoxicants with established acute and chronic effects. Carbofuran, despite its low detection frequency, yielded a notable risk index (0.018) due to its extremely low ADI. Chlorpyrifos, though largely phased out in the EU, remains a frequent contaminant in imported fruit, contributing 0.018 to the total risk. Dimethoate (0.015), indoxacarb (0.014), and tetraconazole (0.010) further illustrate the diversity of high-contribution residues spanning multiple chemical classes and toxicological profiles.

Importantly, many of these substances share overlapping toxicodynamics, especially among insecticides targeting the nervous system. This raises concern about potential cumulative effects through common modes of action, which are not fully captured in additive models. Regulatory bodies such as EFSA have begun to develop cumulative assessment groups (CAGs) to address such scenarios, but implementation remains limited, particularly in low-resource settings or where Codex-based standards predominate [43,44].

Taken together, while the individual hazard indices for adults remain below critical thresholds, the sheer number and variety of substances involved in dietary exposure warrant closer attention. The dominance of neurotoxic insecticides and post-harvest fungicides in the exposure profile—often associated with imported products—suggests specific points of intervention for risk mitigation. These may include stricter import surveillance, alignment with more protective international standards, and prioritization of high-risk substances for monitoring or exclusion.

This complex landscape of low-level exposures, many of which stem from globally traded commodities, highlights the importance of adopting a precautionary approach in national food safety policy. It also invites further research into mixture effects and population-specific vulnerabilities—especially among children or other sensitive groups—which remain outside the scope of the present assessment but are critical to a comprehensive understanding of pesticide risk in Cape Verde.

## 5. Conclusions

This study provides the first comprehensive assessment of pesticide residues in fruits and vegetables marketed in Cape Verde, combining contamination analysis, dietary exposure estimates, and toxicological risk characterization based on real consumption patterns. The results revealed a marked disparity in contamination levels between food groups, with imported fruits being the dominant source of both the number of pesticide residues and the overall dietary exposure. Despite the high detection rate and diversity of compounds—80 active substances, including many not authorized under Codex standards—only one sample exceeded the EU maximum residue limits (MRLs), and the estimated cumulative hazard index for the adult population remained below the threshold of toxicological concern.

However, a relatively small number of substances accounted for the majority of the toxicological burden, most notably omethoate, oxamyl, imazalil, dithiocarbamates, and iprodione. Several of these compounds are banned or severely restricted in the European Union, highlighting persistent asymmetries between Codex-based regulatory standards and more protective frameworks, particularly those of the EU. Although none of the individual exposures exceeded the ADI, the frequent co-occurrence of multiple residues in single food items, including compounds that share common modes of action such as neurotoxicity, underscores the limitations of additive risk models. The current approach does not consider potential synergistic or antagonistic interactions, which could either amplify or mitigate the effects of chemical mixtures.

The analysis also brings to light the heightened vulnerability of children, a group not included in most routine risk assessments. By adjusting for body weight and consumption patterns in children aged 6–11 years, we found that the cumulative hazard index in this age group more than doubles that of adults, approaching the threshold of toxicological concern. Given children’s physiological susceptibility and higher intake relative to body weight, these findings are particularly concerning and suggest the need for age-specific regulatory benchmarks and monitoring efforts.

To ensure food safety in Cape Verde, especially under the pressures of increasing food imports and globalized supply chains, there is a pressing need to strengthen pesticide residue monitoring systems both at the border and within the domestic market. Measures should prioritize the identification and restriction of high-risk compounds, alignment of national regulations with more protective international benchmarks, and the promotion of sustainable local agriculture as a lower-risk alternative. The methodology developed in this study—grounded in real-world consumption and contamination data—offers a scalable framework for other low-resource settings seeking to evaluate the chemical safety of their food supply under a One Health approach.

Future studies should focus on assessing risks in vulnerable populations, characterizing mixture effects through toxicological and mechanistic studies, and evaluating seasonal and long-term trends in exposure. Addressing these challenges is essential for protecting public health and ensuring equitable access to safe food in Cape Verde and similar contexts.

## Figures and Tables

**Figure 1 foods-14-02639-f001:**
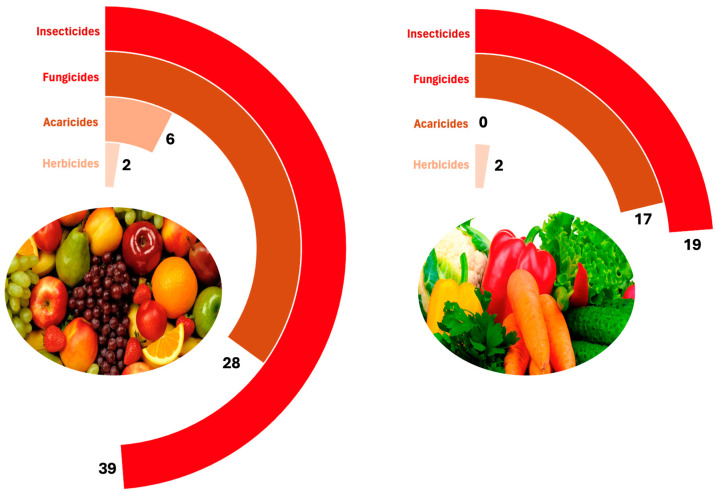
Number of different pesticide active substances detected in fruits (**left**) and vegetables (**right**), classified by type of pesticide.

**Figure 2 foods-14-02639-f002:**
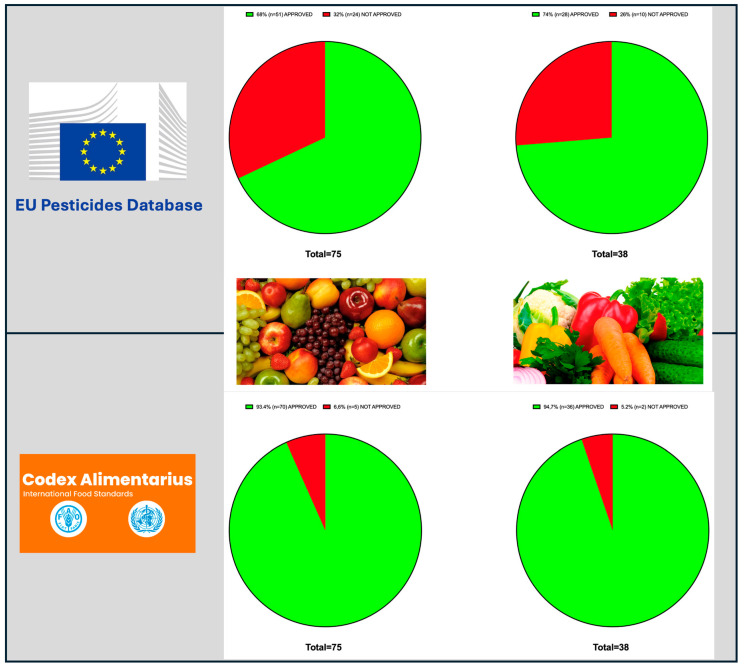
Distribution of active substances detected in fruits and vegetables according to their authorization status under the European Union and considered by the Codex Alimentarius. Each pie chart represents the proportion of authorized and non-authorized substances in each product group by EU and with MRL established by the Codex. MRLs are recommended by the FAO/WHO Joint Meeting on Pesticide Residues (JMPR) and adopted by the Codex Alimentarius; Codex MRLs are often used by countries lacking their own regulatory framework, such as Cape Verde.

**Figure 3 foods-14-02639-f003:**
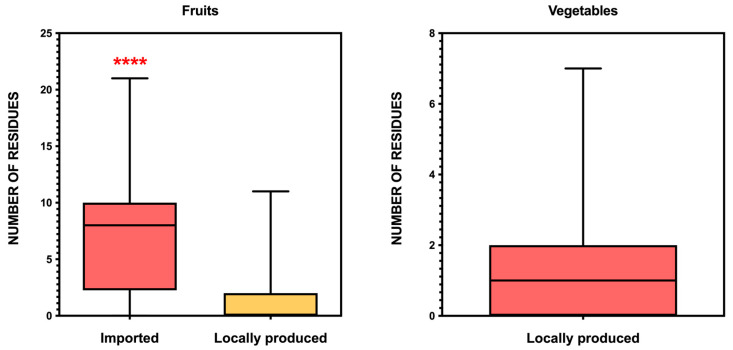
Number of pesticide residues per sample (mean values) detected in each type of fruit and vegetable, classified by origin (imported vs. locally produced). Each bar represents a different type of commodity. This figure allows for the visualization of residue distribution among specific products, complementing the overall average values reported in the text (3.33 residues/sample in imported fruits and 0.89 in locally produced vegetables). For fruits, imported and locally produced items are shown separately. For vegetables, only locally produced samples are included due to the limited number of imported samples (n = 6). **** *p* < 0.0001.

**Figure 4 foods-14-02639-f004:**
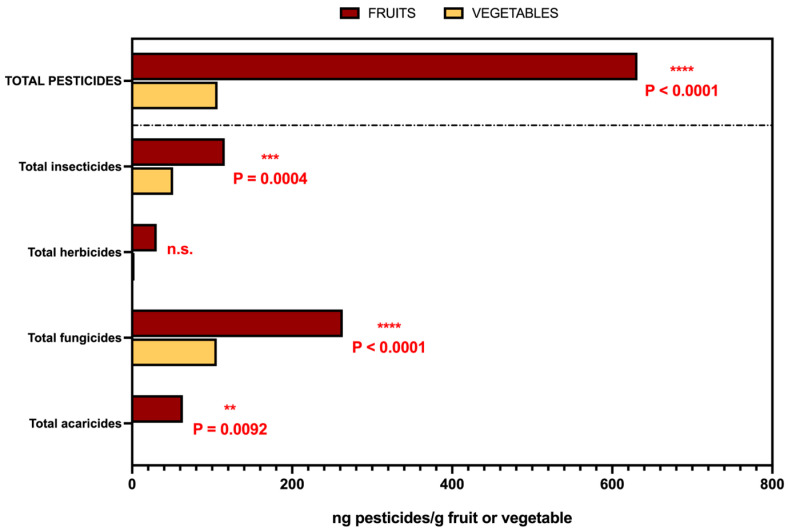
Total concentration of pesticide residues (ng/g) in fruits and vegetables, grouped by pesticide class. Bars represent the total concentrations of acaricides, fungicides, herbicides, and insecticides, as well as the total pesticide load, based on deterministic estimates using medians. This approach reflects the limited granularity of the available consumption data and ensures consistent treatment of censored observations, avoiding artificially inflated variability estimates. For means, SDs, and ranges, see Appendix A. Statistical significance was assessed using the Mann–Whitney U test. Asterisks denote significant differences between food groups (****** *****p***
**< 0.0001**, *** ***p***
**= 0.0004**, ** ***p***
**= 0.0092**), while “n.s.” indicates no significant difference.

**Figure 5 foods-14-02639-f005:**
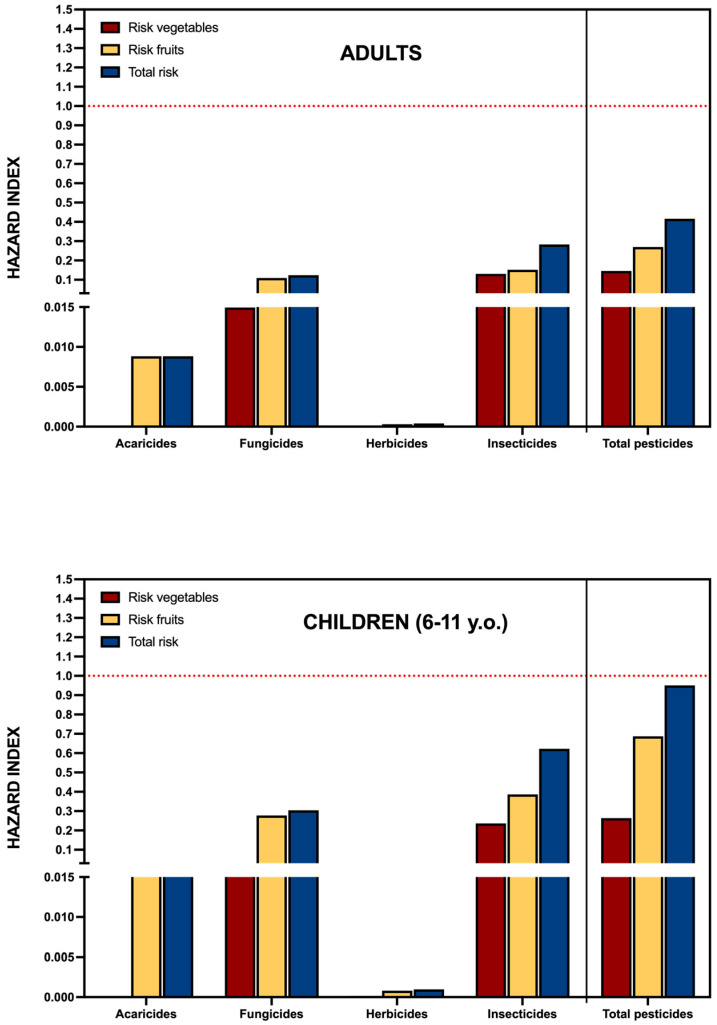
Cumulative hazard index for chronic dietary exposure to pesticide residues in fruits and vegetables among adult (**upper panel**) and child (**lower panel**) consumers in Cape Verde. The index is expressed as the ratio between estimated exposure (mg/kg bw/day) and the Acceptable Daily Intake (ADI) for each active substance, aggregated by pesticide class. Values are presented separately for fruits and vegetables, and as a combined total. All estimates are based on a deterministic approach using median residue concentrations and fixed consumption values from the ENCAVE 2019 survey, in line with standard practice in low- and middle-income countries. Dispersion metrics are provided in Appendix A.

**Table 1 foods-14-02639-t001:** Presence of pesticide residues in the most commonly consumed fruits in Cape Verde. Results are presented as median values and the number of times each pesticide was quantified in each type of fruit.

											Fruits *											
	Apple (n = 22)		Banana (n = 16)		Clementine (n = 15)		Grape (n = 27)		Kiwi (n = 4)		Lemon (n = 7)		Orange (n = 21)		Papaya (n = 20)		Pear (n = 21)		Prune (n = 4)		Watermelon (n = 17)	
Pesticide	N	Median	N	Median	N	Median	N	Median	N	Median	N	Median	N	Median	N	Median	N	Median	N	Median	N	Median
Etoxazole	1	3.4			2	11.6	1	10.0														
Hexythiazox					1	5.6																
Spirodiclofen	5	20.0					2	11.2					1	7.0								
Tebufenpyrad	3	10.0			1	10.0																
Tetradifon	1	10.0																				
**Total acaricides**	8	25.0			8	371.6	1	30.0					10	28.5			1	1.7				
Azoxystrobin					2	530.0							1	60.0			1	8.6	1	10.0		
Boscalid	13	6.7	1	110.0	1	2.3	3	320.0									11	40.0	1	3.8	1	2.6
Carbendazim	2	5.6									1	7.3										
Cyprodinil	1	5.2			1	2.4	2	254.1											1	6.1		
Difenoconazole	13	4.8	1	10.0			1	2.0					1	3.1			9	5.4				
Dimethomorph							1	150.0														
Dinocap							1	7.0														
Diphenylamine	5	6.3					3	2.6									2	2.2				
Dithiocarbamates (sum)	14	43.5			9	80.0	4	43.0					7	67.0	18	900.0	9	140.0				
Fenbuconazole	1	3.3															2	2.2				
Fenhexamid																			1	6.8		
Fludioxonil	13	40.0	1	30.0	4	150.0	2	128.4			1	2710.0	3	2.5			14	5.0	2	16.4		
Fluopyram	7	20.0	1	20.0			2	95.0									13	20.0	1	1.3		
Imazalil	3	330.0			10	1530.0					1	390.0	15	560.0			4	3.9				
Iprodione	1	20.0					2	4.1	2	25.9			1	590.0			7	10.0				
Iprovalicarb													2	2.8								
Kresoxim-methyl							8	6.7									3	20.0				
Metalaxyl							2	6.0														
Metrafenone							4	150.0														
Myclobutanil	5	2.7	1	3.6			2	55.0														
Penconazole							3	7.9							1	6.0						
Prochloraz													2	4.2								
Propiconazole					7	700.0							9	50.0			1	1.7				
Pyraclostrobin	5	20.0	1	50.0	1	7.5	1	20.0			1	40.0	3	40.0			9	10.0				
Pyrimethanil	8	30.0			6	660.0	2	365.0					7	560.0			12	8.8				
Spiroxamine							1	210.0					1	3.2								
Tetraconazole	2	2.2																				
Trifloxystrobin	4	4.8			1	10.0	1	20.0			1	4.6					9	20.0				
**Total fungicides**	21	119.9	1	223.6	13	1191.0	16	26.6	2	25.9	2	1576.0	18	600.0	18	900.0	17	794.2	3	12.7	1	2.6
Chlorprofam	1	6.4							1	3.5			1	2.7								
Paclobutrazol			1	3.5																		
**Total herbicides**	1	6.4	1	3.5					1	3.5			1	2.7								
Acetamiprid																	1	30.0				
Bifenthrin																	1	8.4				
Buprofezin													1	5.8								
Carbofuran													1	4.0								
Chlorantraniliprole	1	30.0					2	45.0														
Chlorpirifos					5	8.0	1	1.7			1	30.0	1	20.0							1	3.3
Chlorpirifos-methyl	3	3.9											2	28.3								
Cyfluthrin	2	8.8																				
Cypermetrin	3	10.0			1	7.5	2	6.5					1	20.0	1	3.5	2	55.0				
Deltamethrin	6	7.8											2	55.0	4	9.8	5	13.5	1	5.3	4	4.6
Dimethoate															4	9.0					2	30.0
Etofenprox	2	1.8			4	5.3	1	250.0					2	111.5					1	2.2		
Fenitrothion																	1	5.2				
Fenpropathrin													2	15.0								
Fenvalerate	1	7.0																				
Esfenvalerate																	1	70.0				
Fipronil																						
Imidacloprid	1	5.1			1	4.0	1	40.0									1	20.0			1	10.0
Indoxacarb							1	9.1					1	97.5					1	2.5		
Lambda-cyhalotrin	9	10.0			4	8.3	1	20.0					2	5.2			4	15.0	1	3.4		
Malathion													2	9.8								
Methidathion													2	4.0								
Omethoate															3	6.9					1	41.3
Oxamyl											1	70.0					1	6.5				
Permethrin											1	7.9			1	40.0						
Phosmet	1	2.4			1	3.4											1	3.8				
Pirimicarb	2	8.2																				
Pirimiphos-methyl	1	3.6															1	4.7				
Profenofos													1	5.3								
Pyridaben					1	80.0																
Pyriproxifen					2	16.4							7	6.9			5	7.3				
Spinosad							1	5.0														
Spirotetramat							1	40.0														
Tebuconazole	11	10.0	1	10.0	2	6.4	10	8.8			1	7.4					17	10.0	1	5.1	1	5.2
Tebufenozide	1	8.1																				
Tetramethrin											1	7.5										
Thau-fluvalinate													1	6.6			1	8.0				
Thiacloprid	5	7.9															1	40.0				
Thiamethoxam							1	5.3									1	8.6				
Thiabendazole	7	10.0	1	250.0	6	79.1					2	130.0	14	175.0			4	4.4				
**Total insecticides**	18	26.0	1	260.0	11	43.8	13	50.0			2	191.4	16	226.2	5	14.7	18	42.6	2	9.3	6	16.2
**Total pesticides**	21	167.9	1	487.0	13	2537.0	19	55.9	2	27.6	2	1767.4	18	1185.4	18	900.0	18	859.2	3	26.5	7	9.1

**Note:** Median values are shown only for pesticide-commodity combinations in which at least one sample yielded a quantifiable residue (≥LOQ). Cells with no quantifiable detections were intentionally left blank to avoid the inclusion of fictitious data. For dietary exposure calculations, however, all samples were included; non-quantified values (<LOQ) were assigned a random value between 0 and the LOQ for the purpose of determining the overall median. * Mango (n = 11) and strawberry (n = 6) samples were also analyzed, and no residues were detected in any of them.

**Table 2 foods-14-02639-t002:** Presence of pesticide residues in the most commonly consumed vegetables in Cape Verde. Results are presented as median values and the number of times each pesticide was quantified in each type of fruit.

													Vegetable											
	Cabbage (n = 34)		Carrot (n = 39)		Coriander (n = 12)		Cucumber (n = 24)		Lettuce (n = 14)		Parsley (n = 9)		Pepper (n = 34)		Potato (n = 11)		Pumpkin (n = 9)		Red Cabbage (n = 41)		Tomato (n = 134)		Zucchini (n = 14)	
	N	Median	N	Median	N	Median	N	Median	N	Median	N	Median	N	Median	N	Median	N	Median	N	Median	N	Median	N	Median
Azoxystrobin							1	540.0																
Boscalid	1	20.0																			1	10.0		
Cyflufenamid													1	6.5										
Difenoconazole													1	40.0										
Dinocap																					1	10.0		
Dithiocarbamates (sum)	26	95.0									2	16.0	11	70.0					38	200.0	27	21.0	1	40.0
Fluopyram	1	40.0																						
Flutriafol	1	6.8																						
Kresoxim-methyl													3	100.0										
Metalaxyl	1	8.0																						
Ofurace									1	2.7														
Penconazole													1	8.1										
Pyraclostrobin			1	20.0																				
Pyrimethanil																					3	2.0		
Spiroxamine																	1	20.0						
Tetraconazole	1	5.8					1	10.0					1	10.0							1	70.0		
Trifloxystrobin	1	3.8																						
**Total fungicides**	**26**	**95.0**	**1**	**20.0**			**2**	**275.0**	**1**	**2.7**	**2**	**16.0**	**14**	**79.1**			**1**	**20.0**	**38**	**200.0**	**30**	**18.5**	**1**	**40.0**
Chlorprofam											1	5.4	2	5.3			1	2.7	1	4.4	2	3.9		
Pendimethalin			1	3.7																				
**Total herbicides**			**1**	**3.7**							**1**	**5.4**	**2**	**5.3**			**1**	**2.7**	**1**	**4.4**	**2**	**3.9**		
Abamectin																					2	6.1		
Chlorpirifos	1	2.8	2	105.0			3	100.0			1	3.2	1	30.0			1	60.0			9	10.0	1	10.0
Chlorpirifos-methyl							1	4.8					3	50.0			1	3.1			4	4.8		
Cypermetrin			1	5.0			1	40.0	1	2.7			1	2.2							6	75.0	3	5.7
Deltamethrin	7	30.0			5	10.7	4	9.1	2	8.3	3	5.3	14	35.0			2	16.5	4	4.7	23	6.5	5	4.3
Dimethoate							4	15.0					7	10.0			2	61.6			5	5.6	3	160.0
Etofenprox																	1	3.7						
Fenitrothion					2	11.5					1	6.6									2	9.2	1	10.0
Fipronil																					1	2.8		
Imidacloprid													4	55.0							6	10.0	2	9.4
Indoxacarb													1	40.0										
Lambda-cyhalotrin	1	120.0											1	70.0										
Omethoate							3	8.8					5	8.2			1	994.3			5	13.6	3	23.2
Permethrin									1	2.1			1	2.8										
Pirimicarb																			2	9.1				
Pyriproxifen																					1	60.0		
Spinosad																					1	9.3		
Tebuconazole	2	11.4							2	3.3			6	320.0					1	3.1	8	3.3	1	2.0
Thiabendazole															1	3.7								
**Total insecticides**	**9**	**30.0**	**3**	**40.0**	**6**	**8.5**	**8**	**45.1**	**4**	**4.8**	**4**	**6.3**	**21**	**70.0**	**1**	**3.7**	**3**	**23.2**	**7**	**5.1**	**43**	**18.1**	**11**	**10.0**
**Total pesticides**	**26**	**124.7**	**4**	**31.9**	**6**	**8.5**	**10**	**45.1**	**5**	**4.8**	**6**	**8.2**	**25**	**130.0**	**1**	**3.7**	**4**	**13.5**	**38**	**202.2**	**65**	**20.0**	**12**	**11.7**

**Note:** Median values are shown only for pesticide–commodity combinations in which at least one sample yielded a quantifiable residue (≥LOQ). Cells with no quantifiable detections were intentionally left blank to avoid the inclusion of fictitious data. For dietary exposure calculations, however, all samples were included; non-quantified values (<LOQ) were assigned a random value between 0 and the LOQ for the purpose of determining the overall median.

**Table 3 foods-14-02639-t003:** Daily exposure to pesticides through fruit and vegetable consumption among the adult population of Cape Verde.

		EXPOSURE ASSESSMENT (µg/adult/day)				
		Acaricides	Fungicides	Herbicides	Insecticides	Total Pesticides
	Consumption (g/day)					
Cabbage	**7.5**	0.00	0.71	0.00	0.23	0.93
Carrot	**17**	0.00	0.34	0.06	0.68	0.54
Coriander	**0.6**	0.00	0.00	0.00	0.01	0.01
Cucumber	**8.2**	0.00	2.26	0.00	0.37	0.37
Lettuce	**2.6**	0.00	0.01	0.00	0.01	0.01
Parsley	**1.4**	0.00	0.02	0.01	0.01	0.01
Pepper	**9.2**	0.00	0.73	0.05	0.64	1.20
Potato	**27.9**	0.00	0.00	0.00	0.10	0.10
Pumpkin	**21.8**	0.00	0.44	0.06	0.51	0.29
Red Cabbage	**7.5**	0.00	1.50	0.03	0.04	1.52
Tomato	**33.5**	0.00	0.62	0.13	0.61	0.67
Zucchini	**14.3**	0.00	0.57	0.00	0.14	0.17
**Total vegetables**	**151.5**	**0.00**	**7.19**	**0.34**	**3.34**	**5.82**
Apple	**45.8**	1.15	5.49	0.29	1.19	7.69
Banana	**72.7**	0.00	16.26	0.25	18.90	35.40
Clementine	**18.3**	6.80	21.80	0.00	0.80	46.43
Grape	**10.7**	0.32	0.28	0.00	0.54	0.60
Kiwi	**18.3**	0.00	0.47	0.06	0.00	0.51
Lemon	**2**	0.00	3.15	0.00	0.38	3.53
Orange	**44.3**	1.26	26.58	0.12	10.02	52.51
Papaya	**32.9**	0.00	29.61	0.00	0.48	29.61
Pear	**29.9**	0.05	23.75	0.00	1.27	25.69
Prune	**18.6**	0.00	0.24	0.00	0.17	0.49
Watermelon	**22.3**	0.00	0.06	0.00	0.36	0.20
**Total fruits**	**315.8**	**9.58**	**127.68**	**0.73**	**34.12**	**202.67**
**Total fruits + vegetables**	**467.3**	**9.58**	**134.87**	**1.07**	**37.46**	**208.49**

**Note:** Values represent median exposures derived from quantified residues only (≥LOQ), based on consumption data from ENCAVE 2019 and residue concentrations measured in the present study. The “Total” corresponds to the sum of medians across all quantified pesticide–commodity combinations within each food category. Residues not detected above the LOQ in any sample were excluded from this table to avoid speculative imputed values. However, in the actual dietary risk assessment, all samples were considered, assigning random values between 0 and the LOQ to non-quantified samples, as detailed in Section 2.4.

## Data Availability

All relevant data have been included either in the main text of the manuscript or in the Appendix A. Any additional data can be obtained from the corresponding author.

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
