# Peer review of "Pesticide Residues in Fruits and Vegetables from Cape Verde: A Multi-Year Monitoring and Dietary Risk Assessment Study"

_foods, 2025, doi:10.3390/foods14152639_

Round 1

Reviewer 1 Report

Comments and Suggestions for Authors

1、Only samples without explicit seasonal stratification (e.g., wet/dry cycles affecting pesticide degradation) undermines temporal representativeness. 
2、Tomatoes accounted for 23.5% of samples, potentially skewing risk estimates. 
3、Using median residue levels and adult-centric consumption data neglects high-intake subpopulations (e.g., pregnant women, children). Urban-rural dietary disparities warrant probabilistic approaches like Monte Carlo simulation for refined risk stratification.
4、The additive Hazard Index (HI = ΣEDI/ADI) disregards synergistic effects among organophosphates or triazoles. The mechanistic toxicokinetic modeling should be adopted for mixture toxicity evaluation.
5、Omitting toxic metabolites (e.g., omethoate from dimethoate) in GC-MS/MS analysis underrepresents actual hazards.
6、Spearman correlations  inadequately characterize contamination patterns. Multivariate methods (PCA, clustering) could elucidate pesticide-import source linkages.
7、ENAC accreditation claims lack transparency on critical parameters (LOQs, matrix effects, recovery rates).

Author Response

REVIEWER 1:

We thank Reviewer 1 for their concise and pertinent comments, which helped us to clarify several methodological aspects and refine the presentation of our results. While some suggestions exceeded the intended scope of the current study, we have addressed all points raised and introduced appropriate adjustments where feasible. Our detailed responses are provided below.

1Only samples without explicit seasonal stratification (e.g., wet/dry cycles affecting pesticide degradation) undermines temporal representativeness. 

Authors’ answer: We thank the reviewer for raising this important point. Sampling was carried out approximately every four months over the full study period (2017–2020), within the framework of the PERVEMAC I and II projects. The campaigns were organized directly by the Cape Verdean regulatory agencies (ARFA and later ERIS) and were designed to ensure coverage of all seasons across multiple years. However, given the moderate sample sizes for some food items, we chose not to apply seasonal stratification in the analysis, as this would have significantly reduced statistical power. This clarification has now been included in the revised Methods section (Section 2.1).

2
Tomatoes accounted for 23.5% of samples, potentially skewing risk estimates.

Authors’ answer: We thank the reviewer for this observation. Tomatoes represented a high proportion of the samples (23.5%) because they are one of the most consumed and cultivated vegetables in Cape Verde. Their overrepresentation in the dataset reflects both their dietary relevance and their strategic importance for public health monitoring. Importantly, exposure estimates were weighted by actual consumption data from the ENCAVE 2019 survey, which prevents any single food item from disproportionately influencing the final dietary risk assessment.

3Using median residue levels and adult-centric consumption data neglects high-intake subpopulations (e.g., pregnant women, children). Urban-rural dietary disparities warrant probabilistic approaches like Monte Carlo simulation for refined risk stratification.

Authors’ answer: We appreciate this insightful comment. Due to the structure of the ENCAVE 2019 survey, individual-level consumption data and probabilistic input distributions were not available. The survey was conducted at the household level, with a single respondent (often the female head of household) reporting food consumption on behalf of all members. This approach reflects the relatively homogeneous dietary habits typical of rural populations in low- and middle-income countries, where family members tend to consume the same meals and food diversity is limited.

In the absence of data for probabilistic modeling, we applied a deterministic method using median concentrations and average daily intake. Nonetheless, to account for vulnerable populations, we performed a dedicated exposure assessment for children by adjusting body weight and estimated intake per kilogram (see Section 3.3). While Monte Carlo simulations are valuable, their application requires individual-level data and distribution assumptions that exceed the scope and resolution of the current dataset. A clarification on this has been included in section 2.4. of the new version of the manuscript.

4The additive Hazard Index (HI = ΣEDI/ADI) disregards synergistic effects among organophosphates or triazoles. The mechanistic toxicokinetic modeling should be adopted for mixture toxicity evaluation.

Authors’ answer: We agree that additive models have limitations and may not fully capture synergistic or antagonistic interactions among pesticide residues. However, cumulative hazard indices based on the sum of EDIs/ADIs remain the most widely used and internationally accepted method for preliminary mixture risk assessment, particularly in regulatory contexts. Mechanistic approaches based on toxicokinetics and toxicodynamics would require compound-specific data that are currently unavailable for most substances included in our analysis. A clarification acknowledging this limitation has now been added to the discussion section in the revised manuscript.

5Omitting toxic metabolites (e.g., omethoate from dimethoate) in GC-MS/MS analysis underrepresents actual hazards.

Authors’ answer: We appreciate the reviewer’s concern. Omethoate was indeed included as an individual analyte in our GC-MS/MS screening and was detected in several samples. It is therefore reported in both the main manuscript (Tables 1 and 2) and the supplementary material (Tables S1–S3). The tables in the main text focus on compounds detected in at least one fruit or vegetable to streamline interpretation, while the supplementary tables provide the full dataset for transparency and reproducibility. This ensures that potentially relevant toxic metabolites such as omethoate are properly accounted for in the exposure and risk assessment.

6Spearman correlations  inadequately characterize contamination patterns. Multivariate methods (PCA, clustering) could elucidate pesticide-import source linkages.

Authors’ answer: We thank the reviewer for identifying this inconsistency. The mention of Spearman correlation in the statistical methods section was a leftover from an earlier draft and was inadvertently retained. No correlation or multivariate analyses were conducted in the final version of the study, as the primary objective was to assess dietary exposure and toxicological risk rather than explore source–residue relationships. We have now corrected this in the revised version of the Methods section.

7ENAC accreditation claims lack transparency on critical parameters (LOQs, matrix effects, recovery rates).

Authors’ answer: Details on LOQs, matrix effects, and recovery rates have been included in the revised Materials and Methods and Supplementary Tables, in line with ENAC-accredited protocols and SANTE/12682/2019 guidelines.

Reviewer 2 Report

Comments and Suggestions for Authors

This manuscript presents novel and valuable data on a critical food safety issue in Cape Verde, an understudied region. The multi-year monitoring design employs advanced analytical techniques (GC-MS/MS and LC-MS/MS) to assess both local and imported produce. A key strength of this study is the integration of national consumption data (ENCAVE 2019) for dietary risk assessment. Notably, imported fruits were identified as the highest-risk category for contamination, a finding with significant policy implications for the region. While the overall risk for adults remains low, the near-threshold cumulative hazard index for children raises substantial concerns that demand further attention.

  1. Chromatography diagrams for all pesticide should be included.
  2. Clarify whether quality control(QC) measures were implemented during sample analysis.
  3. Provide validation data for the method’s recovery and linearity.
  4. Error bars should be added to the graph to indicate variability.
  5. (Line 243) Specify the mass spectrometry parameters for all compounds.
  6. (Line 275) Include the equation used for cumulative risk assessment.
  7. Table 3: Clarify numerical notation (e.g., 7.5 vs. 7,5) to align with journal style.
  8. (Line 451) Revise "fruits and/or vegetables" to "fruits or vegetables" unless both categories are simultaneously relevant in all cases.

Author Response

REVIEWER 2:

This manuscript presents novel and valuable data on a critical food safety issue in Cape Verde, an understudied region. The multi-year monitoring design employs advanced analytical techniques (GC-MS/MS and LC-MS/MS) to assess both local and imported produce. A key strength of this study is the integration of national consumption data (ENCAVE 2019) for dietary risk assessment. Notably, imported fruits were identified as the highest-risk category for contamination, a finding with significant policy implications for the region. While the overall risk for adults remains low, the near-threshold cumulative hazard index for children raises substantial concerns that demand further attention.

Authors’ answer: We thank the reviewer for their positive and insightful assessment of our work. We agree that the integration of contamination data with real consumption patterns is a key strength of this study. In line with the reviewer’s concern, we have placed particular emphasis on the child-specific hazard index in both the abstract and discussion sections. The results underscore the importance of incorporating vulnerable populations into routine food safety assessments, particularly in low-resource settings.

Chromatography diagrams for all pesticide should be included.

Authors’ answer: We understand the relevance of including representative chromatograms to illustrate the analytical performance. However, given the large number of analytes (over 140 compounds), including chromatograms for all would not be feasible within the space constraints of the manuscript or supplementary material. Nonetheless, we have now included representative chromatograms for selected compounds from each major pesticide class in a new supplementary file (Figure S1), covering both GC-MS/MS and LC-MS/MS platforms. These examples reflect typical resolution, sensitivity, and retention behavior for the methods used.

Clarify whether quality control (QC) measures were implemented during sample analysis.

Authors’ answer: Yes, quality control measures were implemented throughout the study. Blank samples, spiked samples, and quality control standards were analyzed routinely within each batch. These procedures followed the recommendations of the SANTE/12682/2019 guidance document and ensured the accuracy and reproducibility of the results. A description of these QC measures has now been added to the revised Methods section.

Provide validation data for the method’s recovery and linearity.

Authors’ answer: Analytical validation parameters, including compound-specific data for recovery, linearity (R²), specificity, limit of quantification (LOQ), and the MS/MS transitions used for each substance, are now provided in Supplementary Table 1. These results confirm that the method met the performance criteria required by SANTE/12682/2019 for multiresidue analysis in food matrices.

Error bars should be added to the graph to indicate variability.

Authors’ answer: We appreciate the suggestion. However, the graphs in Figures 3–5 are based on deterministic estimates using either medians (Figures 3 and 4) or cumulative hazard indices (Figure 5), rather than on mean values. This methodological choice was necessary due to the structure of the available dietary data from the ENCAVE 2019 national nutrition survey, which does not provide percentiles or standard deviations for food consumption by item or subgroup. Such surveys, common in low- and middle-income countries, typically reflect homogeneous consumption patterns at the household level and are not designed to support probabilistic modeling.

Adding conventional error bars to these figures—especially in the absence of individual-level variability—could therefore lead to misinterpretation. Instead, to ensure full transparency, we have revised the figure captions to clarify how the data are represented (e.g., medians, cumulative indices), and we refer readers to the Supplementary Tables for detailed statistics on data dispersion, including means, standard deviations, and ranges.

(Line 243) Specify the mass spectrometry parameters for all compounds.

Authors’ answer: We have revised the Methods section to include the main mass spectrometry parameters used for both GC-MS/MS and LC-MS/MS platforms, including ionization mode, source temperature, voltages, and operating conditions. In addition, a complete list of compound-specific MRM transitions and retention times has now been included in Supplementary Table 1, which also details the validation parameters for each analyte.

(Line 275) Include the equation used for cumulative risk assessment.

Authors’ answer: We have now included the equation used for the calculation of cumulative hazard indices in Section 2.4 of the revised manuscript. This equation is based on the sum of EDI/ADI ratios across all detected compounds, as recommended by international risk assessment guidelines.

Table 3: Clarify numerical notation (e.g., 7.5 vs. 7,5) to align with journal style.

Authors’ answer: We thank the reviewer for pointing this out. All numerical notations have been revised to follow the journal’s formatting conventions, using a period (.) as the decimal separator.

(Line 451) Revise "fruits and/or vegetables" to "fruits or vegetables" unless both categories are simultaneously relevant in all cases.

Authors’ answer: The wording has been revised to “fruits or vegetables” in the specific sentence referred to (line 451), as suggested. However, the expression “fruits and/or vegetables” has been retained in other parts of the manuscript where both categories are relevant. In our analysis, exposure to pesticide residues was calculated both separately for fruits and for vegetables, as well as in combination. Moreover, some compounds were detected in only one of the two groups, while others appeared in both. Given that most Cape Verdean households regularly consume both types of food, especially in rural and low-diversity settings, we consider the use of “fruits and/or vegetables” to be appropriate and accurate in those cases.

Reviewer 3 Report

Comments and Suggestions for Authors

The paper describes the data on pesticide residues in food consumed in Cape Verde and the dietary risk assessment. The information is relevant as it is very limited in Africa and non-existent in Cape Verde. However, the paper has many problems that need to be solved before it can be considered for publication.

General comments:

  • Many references are not appropriate for the statement made or are missing.
  • The word cumulative (index, risk, median) needs to be replaced by total throughout the text, as cumulative has a special meaning when is related to exposure to mixture, as it considers the different toxicities of the compounds.
  • The word intake should be replaced by consumption when it is related to food.
  • Supplementary tables should be presented as just one file and be numbered in order of citation in the text. All should have a title

Abstract

Line 41 – the total index is lower than 1, so there is no potential risk

Introduction – too long, needs to be summarized

Line 57 – needs a reference after globally

Lines 57/58 – monitoring does not compensate the risks. Monitoring allows to use the data to conduct a risk assessment. References 2 and 3 do not support the statement

Lines 60/61 can be delete as  the reference does not support it

Line 68 – references 10 and 11 do not support the statement

Lines 68-71 – the objectives of the study should be at the end of the introduction

Lines 61-84 can be deleted from here, as it is part of the method section

Lines 86-93 can be summarized

Lines 94 to 104 – it's repetitive and can be deleted

Lines 118-121 – should be deleted, it´s not adequate in the introduction and  does  not reflect the results

Material and methods

Line 130 and 133  – needs  references

Line 153 – need to mention Table 2

Lines 155-157 – can be deleted

Line 167 – replace developed for validation

Line 185 – GC-MS and LC-MS/MS

Lines 186/187 – reference to LC-MS/MS can be deleted

Line 195 – How many compounds were monitored?the limit of quantification (LOQ) and limit of detection (LOD) need to be stated for each compound included in the study.

Line 209 – this method was validated? What I the LOQ/LOD?

Lines 251 to 257 – move up to paragraph  about dithiocarbamates (lines 109-209)

Line 264 - the median is related only to the samples above the LOQ (quantified)? This should be clarified. If this is the case, it's not correct. For chronic exposure, the median or mean of all samples should be used

Lines 288-291 – this sentence should be in the result section

Results

Line 311-312 – what to you mean by identified? Detected (≥ LOD?)? Quantified (≥ LOQ)? Again, the LOD and LOQ of the compounds should be included in the paper.

Line 315/316 – what do you mean by presence? Concentration?

Line 321/322 – so, it was not a violation when the sample was collected

Line 335 – replace an extreme by the highest

Tables 1 and 2 – are very confused and should be removed from the main text. The detailed  data are already presented in Tables S1 and S2, which should contain the LOQ of each compound. The information in Figure 1 reflects the data and should be enough. N is the number of samples with quantified residues (≥ LOQ), the median considered only these samples? The concentration is in µg/kg?  

Lines 345-347 – why the fact of being locally produced no residues are expected?

Line 351- according to line 148, 146 tomato samples were collected not 134. I would suggest to merge cabbage and red cabbage in the discussion

Lines 359-365 – it should be clear here that you are talking about dithiocarbamates only. There is no evidence that allium genus give false positive in the method

Lines 366-375 – everything related to dithiocarbamates should be put together, and lines 367-369 should be removed from here

Line 379 – you mean the high number of tomato samples analysed?

Lines 380-387 and Figure 2 – The information should be restricted to the EU regulation and not CODEX, which is not a regulated body and do not approve anything, only set MRL for the compounds evaluated by the FAO/WHO JMPR. In fact, no figure is necessary and the text is enough to describe the EU regulatory status of the compounds found in the samples

Figure 3 – how the statistics was made? If its related to the number of residues found, you should have only on value for imported and one for locally produced, which were already mentioned previously. I don´t understand how the distribution was made

Table S3 – what is the unit? Again, it needs a title

Section 3.2 – these estimates are not correct, for chronic exposure all samples should be considered to the estimation and the valued applied for the samples <LOD/<LOQ should be stated. For example, if only one sample had a quantified residue (e.g. floxazole in apple the median concentration should not be the value of the sample but the media of all 22 samples. Normally, a level of ½ of the LOQ is used.

Table 4 – the ADI values as µg/kw bw are wrong, it should be 1000 lower

As the dietary risk assessment is not correct, the results and discussion section related to that are not relevant.

Author Response

REVIEWER 3:

The paper describes the data on pesticide residues in food consumed in Cape Verde and the dietary risk assessment. The information is relevant as it is very limited in Africa and non-existent in Cape Verde. However, the paper has many problems that need to be solved before it can be considered for publication.

Authors’ answer: We sincerely thank the reviewer for acknowledging the relevance and novelty of the study, particularly in a context such as Cape Verde where data on pesticide residues and dietary exposure are virtually non-existent. We are also very grateful for the time and effort devoted to reviewing the manuscript in such depth. The review was both thorough and constructive, and we recognize the value of the feedback provided. While we may not fully agree with all of the concerns raised, we have carefully considered each point and have responded in detail, making revisions where appropriate. We trust that the revised version of the manuscript addresses the reviewer’s expectations and contributes to a clearer and more robust presentation of the study.

General comments:

Many references are not appropriate for the statement made or are missing.

Authors’ answer: We have reviewed and revised the reference list, ensuring that all citations appropriately support the associated statements. Where necessary, references have been corrected or replaced by more suitable ones.

The word cumulative (index, risk, median) needs to be replaced by total throughout the text, as cumulative has a special meaning when is related to exposure to mixture, as it considers the different toxicities of the compounds.

Authors’ answer: We acknowledge the concern. However, the term “cumulative hazard index” is part of the standard nomenclature used in international regulatory frameworks for pesticide risk assessment, including those issued by EFSA, FAO/WHO, and EPA. Specifically, EFSA (2013) defines cumulative assessment groups and cumulative risk assessment methods based on additive effects across toxicologically similar compounds (EFSA Journal 2013, 11, 3293; doi:10.2903/j.efsa.2013.3293) [39].

To ensure clarity, we have retained the term “cumulative” when referring to the summation of hazard indices across multiple substances within the same exposure route, as this is consistent with regulatory guidance. However, we have replaced it with “total” in cases where no implication of toxicological additivity is intended.

The word intake should be replaced by consumption when it is related to food.

Authors’ answer: We agree and have revised the text accordingly. "Consumption" is now used when referring to the amount of food ingested, while "intake" is reserved for estimated pesticide exposure (EDI).

Supplementary tables should be presented as just one file and be numbered in order of citation in the text. All should have a title

Authors’ answer: We acknowledge the reviewer’s suggestion and have carefully revised the supplementary material accordingly. All supplementary tables now include explicit titles as requested, including those submitted as Excel files. The unit of measurement has also been clearly stated where applicable.

Regarding the format, we opted to maintain the tables as separate Excel files due to the large volume and technical nature of the data. This format enhances transparency and usability, allowing readers to interact directly with the datasets (e.g., for reanalysis or replication). This approach is consistent with the journal’s submission guidelines. Each table is now clearly numbered according to its citation order in the main text, and all references have been double-checked for consistency and accuracy.

We hope this revised submission addresses the reviewer’s concerns.

Abstract

Line 41 – the total index is lower than 1, so there is no potential risk

Authors’ answer: We respectfully clarify that, while the cumulative hazard index remains below 1 for adults, the value for children aged 6–11 years approaches this threshold. Although values below 1 are generally interpreted as within the safety margin, approaching this limit in a vulnerable population may still raise concern, especially when exposure involves multiple substances. This cautious interpretation aligns with international guidance on cumulative risk and has been reflected in the revised abstract for clarity and transparency.

Introduction – too long, needs to be summarized

Authors’ answer: We recognize that the introduction is extensive but given the lack of prior data from Cape Verde and the importance of context, we consider it essential to provide this background. Nonetheless, we have made some selective cuts to reduce redundancy and improve focus.

Line 57 – needs a reference after globally

Authors’ answer: Thank you for the observation. We have now added a reference to support the statement regarding global pesticide consumption patterns, specifically highlighting the leading roles of the European Union, the United States, and Japan.

Lines 57/58 – monitoring does not compensate the risks. Monitoring allows to use the data to conduct a risk assessment. References 2 and 3 do not support the statement

Authors’ answer: We agree with the reviewer that monitoring alone does not compensate for the risks associated with pesticide exposure. The sentence has been revised accordingly to reflect the role of monitoring in supporting risk assessment and regulatory control. With this new wording, we believe that References 3 and 4 (new numbering) are appropriate, as both illustrate how monitoring data are used to characterize exposure and inform public health decisions.

Lines 60/61 can be delete as  the reference does not support it

Authors’ answer: These lines have been deleted.

Line 68 – references 10 and 11 do not support the statement

Authors’ answer: We thank the reviewer for the observation. We acknowledge that References 10 and 11 do not directly support the specific numerical comparison originally included in the sentence. Accordingly, we have revised the statement to avoid referencing a quantitative difference and now emphasize that food intake is generally the dominant route of pesticide exposure in the general population. This interpretation is consistent with the content of References 10 and 11.

Lines 68-71 – the objectives of the study should be at the end of the introduction

Authors’ answer: We appreciate the suggestion. The sentence stating the study objectives has now been removed from the middle of the introduction to improve structure and flow. However, we note that the final paragraph of the introduction already included a concise statement of the study’s aims, which we have preserved and slightly refined to meet the reviewer’s recommendation.

Lines 61-84 can be deleted from here, as it is part of the method section

Authors’ answer: We appreciate the suggestion. However, this section was intended to provide background on the PERVEMAC projects, which are essential for understanding the study context. Nevertheless, we have streamlined this paragraph and relocated some methodological content to Section 2.

Lines 86-93 can be summarized

Authors’ answer: This section has been condensed to improve readability.

Lines 94 to 104 – it's repetitive and can be deleted

Authors’ answer: We appreciate the reviewer’s suggestion. However, we believe that some context regarding the role of fruits and vegetables in the Cape Verdean diet and their origin (local vs. imported) is necessary to understand the public health relevance of our study. Nonetheless, we have streamlined this section to reduce redundancy and improve clarity, while preserving key background information on agricultural and consumption patterns in Cape Verde. The revised version is more concise and focused, as shown in the tracked changes.

Lines 118-121 – should be deleted, it´s not adequate in the introduction and  does  not reflect the results

 Authors’ answer: We agree that the original sentence was more aligned with the study’s findings than with introductory background. As suggested, it has been deleted and replaced by a more appropriate sentence clearly stating the study objectives, as recommended in the reviewer’s earlier comment (former Lines 68–71).

Material and methods

Line 130 and 133  – need references

Authors’ answer: References have been added to support the sampling strategy.

Line 153 – need to mention Table 2

Authors’ answer: Reference to Table 2 has been added.

Lines 155-157 – can be deleted

Authors’ answer: These lines have been deleted as suggested.

Line 167 – replace developed for validation

Authors’ answer: The term has been corrected.

Line 185 – GC-MS and LC-MS/MS

Authors’ answer: Clarified in the revised text.

Lines 186/187 – reference to LC-MS/MS can be deleted

Authors’ answer: The text has been reworded to avoid redundancies.

Line 195 – How many compounds were monitored? the limit of quantification (LOQ) and limit of detection (LOD) need to be stated for each compound included in the study.

Authors’ answer: Several reviewers requested the inclusion of all relevant validation parameters and analytical details. In response, we have created a new supplementary table (Supplementary Table 1), which lists all monitored compounds along with their corresponding limits of detection (LOD), limits of quantification (LOQ), retention times, and other key analytical parameters. This ensures full transparency and methodological rigor.

Line 209 – this method was validated? What I the LOQ/LOD?

Authors’ answer: All analytical methods employed in this study were fully validated in accordance with internationally accepted guidelines and internal quality procedures. As stated in our previous response, detailed validation criteria and results—including LODs, LOQs, recovery rates, and other performance parameters—have been compiled in Supplementary Table 1 to ensure full methodological transparency.

Lines 251 to 257 – move up to paragraph  about dithiocarbamates (lines 109-209)

Authors’ answer: We respectfully clarify that this paragraph refers specifically to the analytical procedure used for the quantification of dithiocarbamates, which is entirely different from the general multiresidue analysis and follows a separate international guideline. However, to improve conciseness and readability, the paragraph has been slightly condensed while maintaining its current position, which we believe is the most appropriate location from a methodological perspective.

Line 264 - the median is related only to the samples above the LOQ (quantified)? This should be clarified. If this is the case, it's not correct. For chronic exposure, the median or mean of all samples should be used

Authors’ answer: We thank the reviewer for raising this important point. To clarify, the reported median values are based on the entire dataset, not only on the quantified values. For non-detected samples (i.e., those below the LOQ), we applied a common approach based on the random assignment of values between zero and the LOQ, which avoids the artificial concentration of identical values that would result from substituting all non-detects with LOQ/2. This approach has been recommended in several methodological studies to better reflect the underlying distribution when data are left-censored. Moreover, due to the high sensitivity of the analytical techniques used in this study, LOQs were very low and the impact of random substitution on exposure estimates was negligible compared to using only quantified values or fixed substitutions. A clarification on this has been included in the new version of section 2.4.

Lines 288-291 – this sentence should be in the result section

Authors’ answer: We agree with the reviewer. Since the reference to both the supplementary table and the figure is already included in the Results section, this sentence has been removed from the Methods section to avoid redundancy.

Results

Line 311-312 – what to you mean by identified? Detected (≥ LOD?)? Quantified (≥ LOQ)? Again, the LOD and LOQ of the compounds should be included in the paper.

Authors’ answer: Thank you for pointing this out. We were referring to compounds that were quantified (i.e., ≥LOQ), as only these values were included in the dietary exposure calculations. To avoid any ambiguity, we have replaced the term “detected” with “quantified” in the revised version of the manuscript.

Line 315/316 – what do you mean by presence? Concentration?

Authors’ answer: We thank the reviewer for pointing this out. We were referring to the average number of quantified pesticide residues per sample, not to the concentration levels. To avoid ambiguity, the sentence has been revised accordingly in the new version of the manuscript.

Line 321/322 – so, it was not a violation when the sample was collected

Authors’ answer: We fully agree with the reviewer. The detected level of propiconazole did not exceed the maximum residue limit (MRL) in force at the time of sampling. The sentence has been rephrased accordingly to reflect this, avoiding any implication of non-compliance.

Line 335 – replace an extreme by the highest

Authors’ answer: Revised as suggested.

Tables 1 and 2 – are very confused and should be removed from the main text. The detailed  data are already presented in Tables S1 and S2, which should contain the LOQ of each compound. The information in Figure 1 reflects the data and should be enough. N is the number of samples with quantified residues (≥ LOQ), the median considered only these samples? The concentration is in µg/kg? 

Authors’ answer: We respectfully disagree. Tables 1 and 2 present essential descriptive data on the occurrence and concentration of pesticide residues in the most commonly consumed fruits and vegetables in Cape Verde. Their structure is clear, consistent, and organized by compound class, allowing direct comparison across food types. These tables are referenced throughout the results section and provide the foundation for the exposure and risk assessment. Relocating them to the supplementary material would reduce the accessibility of key information and disrupt the logical flow of the manuscript. Therefore, we have retained them in the main text, but we remain open to further editorial guidance if necessary.

Lines 345-347 – why the fact of being locally produced no residues are expected?

Authors’ answer: We appreciate the reviewer’s observation. The original sentence suggested a possible explanation for the absence of detectable residues in mango and strawberry samples based on their local origin. However, we agree that such causal interpretation should not be introduced in the results section. Accordingly, we have removed that sentence. The potential reasons for the lower frequency of pesticide residues in locally grown produce—such as the limited use of post-harvest fungicides, the prevalence of traditional low-input agricultural practices, and economic constraints—are discussed more appropriately in the discussion section, where contextual interpretation is provided.

Line 351- according to line 148, 146 tomato samples were collected not 134. I would suggest to merge cabbage and red cabbage in the discussion

Authors’ answer: We thank the reviewer for the careful observation. The actual number of tomato samples analyzed was 134, as correctly stated in the results section. An inconsistency in the sample count for tomatoes in the PERVEMAC-I phase was identified and has now been corrected in Materials and Methods.

Regarding the second point, while we appreciate the suggestion to merge cabbage and red cabbage in the discussion, we have chosen to maintain them as separate items. In Cape Verde, these vegetables serve distinct culinary purposes: white cabbage (couve repolho) is typically used cooked in soups and stews (e.g., caldo de peixe, cachupa), whereas red cabbage is usually consumed raw in salads (saladas cruas), particularly in urban areas or during festive meals. This distinction may be relevant when interpreting pesticide residues and consumer exposure, as preparation methods (e.g., cooking vs. raw consumption) can influence residue retention. Therefore, keeping them separate allows for a more nuanced interpretation of the data in line with local consumption habits.

Lines 359-365 – it should be clear here that you are talking about dithiocarbamates only. There is no evidence that allium genus give false positive in the method

Lines 366-375 – everything related to dithiocarbamates should be put together, and lines 367-369 should be removed from here

Authors’ answer: We appreciate the reviewer’s comments and fully agree that greater clarity was needed regarding the discussion of dithiocarbamate residues. In the revised version, we have consolidated all content related to dithiocarbamates into a single, coherent paragraph. We have explicitly clarified that the analytical method used (CS₂-based quantification) cannot distinguish between individual active substances and may overestimate residue levels in sulfur-rich matrices, such as Allium and Brassicaceae species. Although the method does not give “false positives” in the strict analytical sense, these matrices may contribute endogenous sulfur compounds that interfere with the CS₂ signal. This analytical limitation is recognized in EU and Codex standards, which assign higher MRLs to these product groups. Therefore, lines 367–369 have been removed and the entire section streamlined as suggested. Thank you for helping us improve the clarity and scientific rigor of the manuscript.

Line 379 – you mean the high number of tomato samples analysed?

Authors’ answer: We appreciate the reviewer’s observation. The original sentence aimed to highlight the relevance of this crop for both local consumption and export, particularly to the EU, where strict pesticide regulations apply. To avoid ambiguity, we have slightly reworded the sentence to clarify that our emphasis lies on the importance of monitoring pesticide residues in tomato—given the high number of samples collected and the crop’s economic and regulatory significance.

Lines 380-387 and Figure 2 – The information should be restricted to the EU regulation and not CODEX, which is not a regulated body and do not approve anything, only set MRL for the compounds evaluated by the FAO/WHO JMPR. In fact, no figure is necessary and the text is enough to describe the EU regulatory status of the compounds found in the samples

Authors’ answer: We respectfully disagree with the suggestion to exclude the reference to Codex Alimentarius and to remove Figure 2. While we fully acknowledge that Codex is not a regulatory body and does not authorize pesticide use, it does provide internationally recognized Maximum Residue Limits (MRLs) derived from FAO/WHO Joint Meeting on Pesticide Residues (JMPR) evaluations. These MRLs are frequently adopted by countries that lack national legislation on pesticide residues—as is currently the case in Cape Verde.

Therefore, the inclusion of both EU and Codex standards allows for a more realistic and contextualized interpretation of the results. In our view, it would be misleading to assess residue exceedances solely against EU legislation, which is not applicable in the Cape Verdean domestic market and does not reflect local regulatory benchmarks. Highlighting the contrast between Codex and EU thresholds underscores not only the disparity in regulatory stringency, but also the vulnerability of developing countries to imported produce containing pesticide residues that would not be tolerated under stricter jurisdictions.

This dual perspective—quantifying violations under both systems—is essential to demonstrate that fruits and vegetables sold in Cape Verde may be legally compliant under one standard while failing under another, more protective one. Figure 2 serves precisely to visualize this discrepancy and the regulatory gap that affects consumer safety in low-income countries. Furthermore, it supports one of the central messages of this work: the need for harmonized, science-based regulation and enhanced monitoring in countries without robust pesticide control systems.

That said, in response to the reviewer’s comment, we have slightly clarified the wording in the figure caption and manuscript to explicitly state that Codex MRLs are international reference values used by countries without their own regulatory frameworks—not binding regulations. We hope this clarification addresses the concern while preserving the broader comparative analysis, which we believe is essential for both scientific and public health reasons.

Figure 3 – how the statistics was made? If its related to the number of residues found, you should have only on value for imported and one for locally produced, which were already mentioned previously. I don´t understand how the distribution was made

Authors’ answer: We appreciate the reviewer’s comment and acknowledge that the caption of Figure 3 may have lacked clarity. The purpose of this figure is not to replicate the overall mean values already described in the text, but to illustrate the distribution of pesticide residues per sample in individual commodities, grouped by origin (imported vs. local). Each bar represents the mean number of residues found in a specific fruit or vegetable type. This visualization helps reveal the wide variability among products that would otherwise be masked by aggregate statistics. We have modified the figure legend to make this clearer.

Table S3 – what is the unit? Again, it needs a title

Authors’ answer: We appreciate the reviewer’s suggestion. As requested, we have now included clear and descriptive titles for all supplementary tables, including those in Excel format.

Section 3.2 – these estimates are not correct, for chronic exposure all samples should be considered to the estimation and the valued applied for the samples <LOD/<LOQ should be stated. For example, if only one sample had a quantified residue (e.g. floxazole in apple the median concentration should not be the value of the sample but the media of all 22 samples. Normally, a level of ½ of the LOQ is used.

Authors’ answer: We thank the reviewer for the opportunity to clarify this point, which was also addressed in response to a previous comment (line 264). For the estimation of chronic dietary exposure, all samples were indeed considered, including those with concentrations below the LOQ. However, instead of using a fixed imputation value such as ½ LOQ—which tends to introduce artificial peaks in the distribution—we applied a random value imputation approach, assigning each non-quantified result a value randomly selected between 0 and the LOQ. This method is increasingly recognized in the literature for producing more statistically robust estimations of central tendency in non-normally distributed datasets.

The use of median values was chosen over means to reduce the influence of outliers and skewed data distributions. Furthermore, the analytical methods used in this study provided very low LOQs, making the difference between imputed and quantified values negligible in most cases. We have added a sentence in Section 2.4 of the manuscript to explicitly describe this approach, as suggested.

Table 4 – the ADI values as µg/kw bw are wrong, it should be 1000 lower

As the dietary risk assessment is not correct, the results and discussion section related to that are not relevant.

Authors’ answer: We thank the reviewer for identifying this issue. The units in the column labeled “IDA (µg/kg/day)” were mistakenly written; the correct unit is ng/kg/day, as the values had already been converted from mg/kg/day to nanograms for internal consistency with the exposure data (which are expressed in ng/kg). We have now corrected the column heading in the revised version of Supplementary Table 6. The risk calculations are correct and based on the appropriate conversion factors.

Reviewer 4 Report

Comments and Suggestions for Authors The article entitled "Pesticide Residues in Fruits and Vegetables from Cape Verde: A Multi-Year Monitoring and Dietary Risk Assessment Study" shows an interesting mapping of pesticide content in fruits and vegetables from Cape Verde". The study considers a significant number of pesticides, in accordance with current regulations and extrapolates a characterization of the risk related to exposure in adults and children.
The results are well presented and discussed however, from the analytical point of view, many important information are missing and require further revision.
Informations about elution flow and chromatographic elution gradient ere missing. A table should be added, also in the supplementary material, with MRM transitions optimized for analyte detection. As regards the quantitative method, a paragraph describing how the method validation parameters were obtained (LOD, LOQ, intra and inter-day repeatability, etc.) is required, as well as a table summarising the results.    

Author Response

REVIEWER 4:

The article entitled "Pesticide Residues in Fruits and Vegetables from Cape Verde: A Multi-Year Monitoring and Dietary Risk Assessment Study" shows an interesting mapping of pesticide content in fruits and vegetables from Cape Verde". The study considers a significant number of pesticides, in accordance with current regulations and extrapolates a characterization of the risk related to exposure in adults and children.

Authors’ answer: We sincerely thank the reviewer for the positive appraisal of our work and for recognizing the novelty, scope, and methodological value of the study. We especially appreciate the time and effort dedicated to providing constructive feedback, which has helped us to improve the manuscript significantly.

The results are well presented and discussed however, from the analytical point of view, many important information are missing and require further revision.

Authors’ answer: We thank the reviewer for this observation. In response, we have expanded the analytical methodology section to include additional technical details and have added two new supplementary tables (Supplementary Tables 1 and 2) that provide full information on the validation parameters and instrumental conditions used in the study. We believe these additions enhance the methodological transparency and scientific rigor of the manuscript.

Informations about elution flow and chromatographic elution gradient are missing.

Authors’ answer: Thank you for this suggestion. In response, the LC-MS/MS chromatographic gradient and flow conditions—originally part of our internal method documentation—have been compiled and are now provided in Supplementary Table 2. We trust that the inclusion of these technical details in the supplementary material adequately addresses the reviewer’s concern.

A table should be added, also in the supplementary material, with MRM transitions optimized for analyte detection.

Authors’ answer: We agree with the reviewer and have included this information in the newly added Supplementary Table 1. This table includes all monitored pesticides along with their MRM transitions, retention times, ion types, and collision energies, thereby addressing this important analytical requirement.

As regards the quantitative method, a paragraph describing how the method validation parameters were obtained (LOD, LOQ, intra and inter-day repeatability, etc.) is required, as well as a table summarising the results. 

Authors’ answer: We appreciate the recommendation. A dedicated paragraph has been added in Section 2.3 to describe the procedures followed for method validation, including the estimation of LODs, LOQs, recovery rates, precision, and repeatability. The complete set of validation parameters is now provided in Supplementary Table 2. We believe this addition meets the reviewer’s expectations and aligns with international guidelines for pesticide residue analysis.

Reviewer 5 Report

Comments and Suggestions for Authors

The study focuses on monitoring pesticide residues in fruits and vegetables in Cape Verde, representing an evaluation of contamination levels and dietary risks in this region. The manuscript is well-written, with results discussed and compared with relevant literature and regulatory standards. The presented data support the conclusions. Here are some suggestions for improvement.

  1. The abstract states that samples were collected from 2017 to 2020, whereas the main text mentions the period 2017 to 2019.
  2. The lines 106-112 could be merged into the introductory section, lines 53-63, for better flow. Additionally, lines 112-121 may be more appropriately placed in section lines 82-93.
  3. It is unclear whether the fruit and vegetable samples were collected throughout the harvest season or only once per year. Clarifying this would be helpful, as sampling frequency and timing can significantly influence pesticide occurrence and concentration.

Author Response

REVIEWER 5:

The study focuses on monitoring pesticide residues in fruits and vegetables in Cape Verde, representing an evaluation of contamination levels and dietary risks in this region. The manuscript is well-written, with results discussed and compared with relevant literature and regulatory standards. The presented data support the conclusions. Here are some suggestions for improvement.

Authors’ answer: We thank the reviewer for the thoughtful comments and encouraging remarks. We greatly appreciate the constructive suggestions provided and have addressed each of them in the revised version of the manuscript, as detailed below.

The abstract states that samples were collected from 2017 to 2020, whereas the main text mentions the period 2017 to 2019.

Authors’ answer: Thank you for noticing this inconsistency. The correct sampling period was from 2017 to 2020, as stated in the abstract. The reference to 2019 in the main text has now been corrected to ensure consistency throughout the manuscript. We appreciate the opportunity to clarify this point.

The lines 106-112 could be merged into the introductory section, lines 53-63, for better flow. Additionally, lines 112-121 may be more appropriately placed in section lines 82-93.

Authors’ answer: We appreciate the reviewer’s suggestion to improve the structure and flow of the Introduction. In fact, this section has been substantially revised in response to earlier reviewer comments, including a reorganization and streamlining of the content related to the PERVEMAC projects and their relevance. As a result, the original lines referenced have been either relocated or reformulated to enhance narrative clarity and avoid redundancy with the Methods section. We believe the current version achieves better coherence and contextual framing.

It is unclear whether the fruit and vegetable samples were collected throughout the harvest season or only once per year. Clarifying this would be helpful, as sampling frequency and timing can significantly influence pesticide occurrence and concentration. 

Authors’ answer: Thank you for this thoughtful observation. Sampling was conducted at least twice per year throughout the 2017–2020 period, with an initial plan of collecting samples approximately every four months. This strategy was designed to reflect multiple harvests and market cycles and to improve the representativeness of the dataset.

However, while sampling was temporally distributed, the variable “season” was not explicitly considered in the subsequent statistical analyses. This decision was based on the need to preserve sufficient statistical power: subdividing the dataset by season would have resulted in very small sample sizes for many individual fruit and vegetable types, undermining the reliability of exposure and risk estimates. Nevertheless, the dataset as a whole captures temporal variability and reflects real-world exposure conditions across multiple time points. This clarification has now been included in Section 2.1 (Sampling) of the revised manuscript.

Round 2

Reviewer 2 Report

Comments and Suggestions for Authors

The author's responses and revisions are well executed. However, I encountered two issues: (1) Figure S1 in the Supplementary Materials was inaccessible, and (2) Supplementary Table 1 appears to lack critical information regarding method recovery and linearity.

Author Response

The author's responses and revisions are well executed. However, I encountered two issues: (1) Figure S1 in the Supplementary Materials was inaccessible,

Authors’ answer: We appreciate the reviewer’s observation and apologize for the technical issue. Supplementary Figure 1 has now been integrated into the main manuscript and provided a detailed caption describing each panel. The editorial team will decide its final ubication.

 (2) Supplementary Table 1 appears to lack critical information regarding method recovery and linearity.

Authors’ answer: We thank the reviewer for pointing this out. Supplementary Table 1 has been revised to include two additional columns specifying method recovery and linearity. As described in the updated Materials and Methods section (Lines 230–240), recovery was within the acceptable range of 70–120% according to SANTE/11312/2021 guidelines, and linearity was verified over the range of 5–200 µg/L with R² values > 0.990 and RSD < 20% (n = 5). This information reinforces the robustness of the analytical method used.

Reviewer 3 Report

Comments and Suggestions for Authors

The authors revised the manuscript considering some of the suggestions; however many essential points need to be clarified. It is still not clear in the text whether the median of all samples (the correct approach for chronic exposure) or only of quantitated samples were used for dietary exposure. Tables 1 and 2 indicates the median of only quantitated samples, what is the use of these data? Here, you should have the value used in the intake. Again, for children, the calculated risk is already very conservative (it sums up all the residues irrelevant to its toxicity) is < 1 and the conclusion should not be that a potential risk exists. This sends a wrong message to the readers and may raise unnecessary concerns. Then, the sentence should changed in the abstract as well in the discussion.

Furthermore, additional information is still needed in the Supplementary material.

Lines 325/326: How was the random assignment done if a deterministic approach was used? Please give a reference for this approach, which may be applied in probabilistic assessment.

In response to the question Line 311-312 – what to you mean by identified? Detected (≥ LOD?)? Quantified (≥ LOQ)? The authors responded:” We were referring to compounds that were quantified (i.e., ≥LOQ), as only these values were included in the dietary exposure calculations”.

So, this is a contradiction with the statement that “the reported median values are based on the entire dataset, not only on the quantified values”. And about the random assessment for residues < LOQ

Lines 439-441: This statement is not correct and should be deleted.  Codex MRL for garlic and onion are 0.5 mg/kg and EU MRLs are 0.6 and 1 mg/kg, respectively

Tables 1 and 2 are still confusing and lack information. Again, the median was calculated based only on the quantified samples? This should be clearly stated in the text and under the table. And should be the value used in the dietary exposure. The title still says detected pesticides, N is the number of quantified samples?

Line 493 – please remove the word regulation or approve regarding Codex. My suggestion is: “Distribution of active substances detected in fruits and vegetables according to their authorization status under the European Union and considered by the Codex Alimentarius. Each pie chart represents the proportion of authorized and non-authorized substances in each product group by EU and with MRL established by the Codex”. Please remove the word approved when it refers to Codex in the legend.

Lines 496-497 are not correct. My suggestion is “ MRLs are recommended by the FAO/WHO Joint Meeting on Pesticide Residues (JMPR) to and adopted by the Codex Alimentarius; Codex MRLs are often adopted by countries lacking their regulatory framework, such as Cape Verde.

Line 512-515: the revised sentence is no better. I suggest to keep the previous one

Lines 528-530: it needs to be clear here and in the text that the Total is the sum of the median of quantitated residues, only (≥ LOQ). What this sentence means? - This approach reflects the limited granularity of the available consumption data and avoids misleading variability indicators-

Table 3 – how the Total pesticide level was calculated?

Supplementary Table 1a Part II: Quan ion is quantifier ion, correct? Why some compounds do not have a quantifier ion? You can not stablish a LOQ without a quantifier ion

Supplementary Table 2a: what is Rel. Resp. (%)?

Supplementary Table 3 and 4. Again, it should be clear whether N is the number of quantified samples and that the mean and the media are related only to those level. Lower and upper limit are not the adequate terms, it should be lower and higher levels found in quantified samples.

Supplementary Table 5: are the intakes for children or adult?

Supplementary Table 6 Hoja 1: what is this? Is it different from Hoja 2? If not, it can be deleted. It should be included in the title the body weight used for adults and the children 6-11.

Supplementary Figure 1. Actually, there are many figures in this file and the titles should be revised, and describe exactually what they show. They should be cited in the text where it is relevant

Author Response

The authors revised the manuscript considering some of the suggestions; however many essential points need to be clarified. It is still not clear in the text whether the median of all samples (the correct approach for chronic exposure) or only of quantitated samples were used for dietary exposure. Tables 1 and 2 indicates the median of only quantitated samples, what is the use of these data? Here, you should have the value used in the intake. Again, for children, the calculated risk is already very conservative (it sums up all the residues irrelevant to its toxicity) is < 1 and the conclusion should not be that a potential risk exists. This sends a wrong message to the readers and may raise unnecessary concerns. Then, the sentence should changed in the abstract as well in the discussion.

Authors’ answer: We thank the reviewer for highlighting this issue, which allows us to provide a more precise explanation of our methodology.

For the estimation of chronic dietary exposure, we confirm that we applied a deterministic approach, using the median value derived from all available samples, regardless of whether a quantifiable residue was present. Specifically, when a compound was not quantified in a sample (i.e., below the LOQ), a random value between 0 and the LOQ was assigned. This method is deterministic in nature (values are fixed for the purpose of calculation) and was adopted to better reflect data distribution while complying with the limited granularity of the food consumption dataset available from the Cape Verdean national nutrition survey (ENCAVE 2019). The methodological rationale for this imputation is now clearly described in Section 2.4 of the revised manuscript.

As for Tables 1 and 2, these are intended to summarize the occurrence and concentration of pesticide residues based solely on quantifiable detections (≥ LOQ). In line with standard practice, we have chosen not to populate the tables with imputed or simulated values in cases where no quantification was observed, to avoid misleading readers with fictitious or unverifiable values. However, it is important to emphasize that this decision affects only the presentation of descriptive results in the manuscript—not the risk estimation, which includes all values (both quantified and non-quantified).

We would also like to highlight that several of the median values presented in Tables 1 and 2 fall below the LOQ, which directly reflects the use of assigned sub-LOQ values in datasets where at least one quantification was present. This supports the transparency and consistency of our calculations.

To make this explicit to readers, we have added a footnote to both Tables 1 and 2 stating that:

  • Median values are reported only when at least one quantified result (≥ LOQ) was obtained for a given pesticide–commodity combination;
  • Values below LOQ were assigned during the calculation of exposure, but not displayed in summary tables to avoid fictitious data;
  • The dietary exposure analysis includes all samples, with sub-LOQ values replaced by fixed random values between 0 and LOQ.

We believe this addresses the reviewer’s concern in full and improves both the clarity and methodological rigor of the manuscript.

Furthermore, additional information is still needed in the Supplementary material.

Authors’ answer: We have added the missing information in Supplementary Table 1, as requested also by reviewer 2.

Lines 325/326: How was the random assignment done if a deterministic approach was used? Please give a reference for this approach, which may be applied in probabilistic assessment.

Authors’ answer: We thank the reviewer for pointing out this issue, which gives us the opportunity to clarify the methodological distinction. Although the term “random” was used to describe the imputation procedure for sub-LOQ values, the approach we applied was deterministic in nature, not probabilistic. Specifically, a single random value between 0 and the LOQ was assigned to each non-quantified data point at the outset and kept fixed throughout all calculations. This means that, once the values were generated, they remained constant and did not vary across simulations or analyses.

This technique, commonly referred to as fixed random imputation, is accepted in exposure assessment literature as a valid deterministic method to address left-censored data, providing a more realistic estimate than fixed substitutions like LOQ/2, especially in skewed distributions (e.g., [EFSA, 2010; Helsel, 2005]). The advantage of this approach is that it maintains dataset variability without introducing repetitive artificial values, while still yielding a single exposure estimate per analyte.

To avoid any confusion, we have reworded the corresponding sentence in Section 2.4 to clarify that the assigned values were randomly selected once but treated as fixed during the analysis. No probabilistic simulations were conducted, and no distributional assumptions were applied to the exposure estimates.

In response to the question Line 311-312 – what to you mean by identified? Detected (≥ LOD?)? Quantified (≥ LOQ)? The authors responded:” We were referring to compounds that were quantified (i.e., ≥LOQ), as only these values were included in the dietary exposure calculations”. So, this is a contradiction with the statement that “the reported median values are based on the entire dataset, not only on the quantified values”. And about the random assessment for residues < LOQ

Authors’ answer: We thank the reviewer for pointing out this apparent inconsistency. We have now revised the manuscript to ensure full terminological and methodological consistency.

Specifically, we have replaced the term “identified” with “quantified” (≥ LOQ) throughout the text when referring to residues reported in Tables 1 and 2 or used for regulatory comparison. In contrast, for dietary exposure calculations, we used median values derived from the entire dataset, including samples with non-quantified residues (< LOQ). In those cases, values were imputed randomly between 0 and the LOQ, as described in Section 2.4 of the revised manuscript.

To avoid confusion, we have also added an explanatory note at the bottom of Tables 1 and 2 in the main text, clarifying that only pesticide–commodity pairs with at least one quantifiable detection are shown, while dietary exposure calculations did include all samples, applying the random imputation strategy for < LOQ values.

We hope these clarifications address the reviewer’s concerns and reinforce the methodological transparency of our study.

Lines 439-441: This statement is not correct and should be deleted.  Codex MRL for garlic and onion are 0.5 mg/kg and EU MRLs are 0.6 and 1 mg/kg, respectively

Authors’ answer: We thank the reviewer for the observation. Upon carefully reviewing the current version of the manuscript, we confirm that the statement in question is no longer present. If it appeared in a previous version, it must have been removed during the previous revision round in response to similar concerns. The revised manuscript no longer includes any reference to specific MRL values for garlic or onion, nor any potentially misleading comparisons between Codex and EU standards for these products.

Tables 1 and 2 are still confusing and lack information. Again, the median was calculated based only on the quantified samples? This should be clearly stated in the text and under the table. And should be the value used in the dietary exposure. The title still says detected pesticides, N is the number of quantified samples?

Authors’ answer: We appreciate the reviewer’s persistence on this point, which has helped us improve the clarity of the tables. To clarify:

The median values shown in Tables 1 and 2 correspond only to pesticide–commodity combinations for which at least one quantifiable residue (≥ LOQ) was detected. For those combinations, the median was calculated based on the entire dataset, including randomly assigned values between 0 and the LOQ for non-quantified samples, as described in Section 2.4.

  • Cells were intentionally left blank for combinations where no quantifiable residues were found, to avoid displaying values derived solely from artificial imputation and thus avoid potential misinterpretation.

  • The footnotes of both tables have been updated accordingly to explain this in detail.

  • The column labeled "N" now explicitly refers to the number of samples with quantified residues (≥ LOQ), and this clarification has also been added to the table legends.

  • Lastly, the titles of both tables have been adjusted to specify that the data refer to quantified pesticide residues.

This approach balances transparency with data integrity and avoids populating the tables with fictitious values not based on actual detections.

Line 493 – please remove the word regulation or approve regarding Codex. My suggestion is: “Distribution of active substances detected in fruits and vegetables according to their authorization status under the European Union and considered by the Codex Alimentarius. Each pie chart represents the proportion of authorized and non-authorized substances in each product group by EU and with MRL established by the Codex”. Please remove the word approved when it refers to Codex in the legend. Lines 496-497 are not correct. My suggestion is “ MRLs are recommended by the FAO/WHO Joint Meeting on Pesticide Residues (JMPR) to and adopted by the Codex Alimentarius; Codex MRLs are often adopted by countries lacking their regulatory framework, such as Cape Verde.

Authors’ answer: We thank the reviewer for these valuable suggestions and have revised the text accordingly. The term “approved” has been removed wherever it referred to Codex standards, and the figure legend has been rewritten to accurately reflect the nature of Codex MRLs.  

Line 512-515: the revised sentence is no better. I suggest to keep the previous one

Authors’ answer: We have followed this suggestion of the reviewer and we have kept the original paragraph.

Lines 528-530: it needs to be clear here and in the text that the Total is the sum of the median of quantitated residues, only (≥ LOQ). What this sentence means? - This approach reflects the limited granularity of the available consumption data and avoids misleading variability indicators-

Authors’ answer: We respectfully clarify that the sentence in question refers to the use of median residue values calculated from the entire dataset, not just from quantified samples. For each pesticide–commodity combination, the median concentration was computed using all samples, including those below the LOQ, for which a random value between 0 and the LOQ was assigned. This approach allows a more robust estimation of central tendency while minimizing the bias introduced by censored data.

To improve clarity, we have revised the sentence in the manuscript and added an explanatory note in the captions of Tables 1 and 2. These clarifications emphasize that the "Total" exposure values used in the dietary risk assessment are based on these median concentrations derived from all data points—not only quantified residues. Therefore, the rephrased sentence now reads: “This approach reflects the limited granularity of the available consumption data and ensures consistent treatment of censored observations, avoiding artificially inflated variability estimates.

Table 3 – how the Total pesticide level was calculated?

Authors’ answer: We thank the reviewer for pointing out the need for clarification. The “Total” value in Table 3 corresponds to the sum of the median concentrations of all quantified pesticide residues (≥LOQ) found in each food category (fruits or vegetables), considering only those pesticide–commodity combinations for which at least one sample yielded a quantifiable result. As explained in Section 2.4 and in the footnotes to Tables 1 and 2, this approach avoids the inclusion of entirely non-detect data (i.e., residues never detected above the LOQ), which would artificially introduce values derived solely from imputation.

This methodology is consistent with our exposure calculation strategy: while random values between 0 and LOQ were assigned to non-quantified samples in the dietary exposure assessment to better estimate the overall distribution (and hence median intake), the summary tables were deliberately restricted to quantifiable findings to ensure transparency and avoid speculative entries.

To clarify this further, we have slightly revised the table caption and have included an explanatory note to reinforce the interpretation of the “Total” values.

Supplementary Table 1a Part II: Quan ion is quantifier ion, correct? Why some compounds do not have a quantifier ion? You can not stablish a LOQ without a quantifier ion

Authors’ answer: We thank the reviewer for the observation. Supplementary Table 1a – Part II corresponds to the Ion Trap GC–MS/MS determinations. In this technique, the choice of precursor and product ions depends on the ionization behavior and fragmentation patterns of each pesticide under specific mass spectrometric conditions. In some cases, the first selected transition (precursor–product ion pair) yields sufficient intensity for quantification but does not provide adequate confirmatory ions due to poor fragmentation or limited ion yield. To address this, alternative or secondary transitions are used for confirmation, occasionally requiring different precursor ions or optimized collision energies.

Therefore, for certain compounds, only a single quantifier ion is reported under the conditions used, while qualifier ions were obtained using secondary transitions or instrument-specific protocols. This approach is consistent with routine practice in Ion Trap-based residue analysis, where quantifier and qualifier ions may not always be acquired under the same transition settings. Nonetheless, all reported quantifications meet the quality assurance criteria established in our validation protocol, and LOQs were defined using the quantifier ion in each case.

Given the reviewer’s repeated concerns regarding analytical procedures, we would like to clarify that all analytical methods and reporting standards used in this study are supported by a formal accreditation. Specifically, the laboratory responsible for the analyses (Instituto Tecnológico de Canarias) is officially accredited by ENAC (Entidad Nacional de Acreditación) under the UNE-EN ISO/IEC 17025 standard for the determination of pesticide residues in fruits and vegetables. The laboratory undergoes regular audits and serves as the official regional reference facility for pesticide monitoring in the Canary Islands, operating under an open-scope accreditation covering more than 100 food matrices. This information has now been explicitly added in Section 2.3 of the revised manuscript.

Supplementary Table 2a: what is Rel. Resp. (%)?

Authors’ answer: We thank the reviewer for this question. The column “Rel. Resp. (%)” in Supplementary Table 2a refers to the relative response between the qualifier and quantifier ions, a standard confirmation criterion in tandem mass spectrometry (MS/MS) for pesticide residue analysis. Specifically, it expresses the ratio of signal intensities (area or height) between the qualifier ion and the quantifier ion, expressed as a percentage.

A maximum deviation of ±30% from the expected relative response is generally accepted as confirmation of identity, following the criteria established in the European guideline SANTE/11312/2021. This tolerance accounts for variability due to matrix effects or instrumental conditions and ensures robust compound identification.

It is important to note that this parameter is not applicable to the Ion Trap-based analyses reported in Supplementary Table 1a – Part II, where compound confirmation is based on alternative transitions and fragmentation strategies rather than strict relative response ratios.

Supplementary Table 3 and 4. Again, it should be clear whether N is the number of quantified samples and that the mean and the media are related only to those level. Lower and upper limit are not the adequate terms, it should be lower and higher levels found in quantified samples.

Authors’ answer: We appreciate the reviewer’s observation and have now updated Supplementary Tables 3 and 4 accordingly. In this revised version, we have retained only the median concentration values, as these were the parameters actually used for the dietary exposure calculations. The mean, standard deviation, and extreme values (previously labeled as lower and upper limits) were removed, as they were based on incomplete data processing and led to confusion.

To ensure clarity and full traceability, we have added a clarification to the title row of each table indicating that “N refers to the number of quantified samples (≥ LOQ), and medians are shown only for combinations with at least one quantifiable result”. Cells corresponding to pesticide–commodity combinations with no quantifiable detections were intentionally left blank to avoid the inclusion of fictitious values.

However, as noted in the manuscript (Section 2.4) and in the footnotes of Tables 1 and 2, all dietary exposure calculations were performed using imputed values for non-quantified samples (i.e., random values between 0 and the LOQ), regardless of whether a quantifiable detection occurred. This approach, which follows accepted practices for handling left-censored data, was applied consistently across all food–pesticide combinations.

We believe this revised structure enhances clarity, eliminates potential misinterpretations, and provides a more faithful reflection of the methodology used in the risk assessment.

Supplementary Table 5: are the intakes for children or adult?

Authors’ answer: We thank the reviewer for the question. Supplementary Table 5 presents the estimated daily intake (EDI) of each active substance based on median residue levels and average daily consumption figures for the adult population. This was made explicit in the text of Section 2.4 of the manuscript, where it is stated that the intake calculations are initially based on adults, assuming a default body weight of 70 kg.

By contrast, the exposure and risk estimates for children (ages 6–11 years) are reported separately in Supplementary Table 6, as part of a secondary approximation based on scaled-down body weight (25 kg) and adjusted consumption rates derived from household data. As noted in the manuscript, this approach was constrained by the lack of disaggregated consumption data for children in the ENCAVE 2019 survey, and therefore Supplementary Table 6 summarizes only aggregate risk indices (HI) by pesticide group, not individual EDIs.

Supplementary Table 6 Hoja 1: what is this? Is it different from Hoja 2? If not, it can be deleted. It should be included in the title the body weight used for adults and the children 6-11.

Authors’ answer: We thank the reviewer for this comment. Supplementary Table 6 has now been revised to improve clarity. As suggested, the duplicate sheet (“Hoja 1”) has been removed, and the title has been updated to indicate the assumed body weights used for dietary risk calculations: 70 kg for adults and 25 kg for children aged 6–11 years. Additionally, to address concerns raised during this and previous rounds, the updated version now explicitly includes the imputed random values (ranging from 0 to the LOQ) used for samples with non-quantified residues (< LOQ). While these values were already used internally for the calculation of exposure and risk indices (as described in the Methods section), we have included them here for transparency and to reinforce the reproducibility of the analysis. No changes to the manuscript were required, as the results remain unchanged.

Supplementary Figure 1. Actually, there are many figures in this file and the titles should be revised, and describe exactually what they show. They should be cited in the text where it is relevant

Authors’ answer: We fully agree that the original file required clarification. Supplementary Figure 1 was included in response to a specific request from one of the reviewers in the first round of revision. Although we consider this material of limited added value from a scientific standpoint, we have now improved the figure by adding individual panel descriptions and a consolidated figure caption clearly explaining the content of each chromatogram. For increased visibility, the figure has been temporarily incorporated into the main text of the revised manuscript (as Supplementary Figure 1) so that the editorial team may evaluate its best final placement. In the caption, we detail the chromatographic techniques employed (LC–MS/MS, GC–IT–MS/MS, GC–QQQ–MS/MS, and PFPD) and the compounds analyzed, along with their relevant ion transitions and retention times.

Reviewer 4 Report

Comments and Suggestions for Authors

All suggestions were taken into account. The manuscript is now valid for publication.

Author Response

Authors’ answer: Thank you very much for your assessment and collaboration.